# Temporal Dynamics of Gene Expression and Metabolic Rewiring in Wild Barley (*Hordeum spontaneum*) Under Salt Stress

**DOI:** 10.3390/ijms27010358

**Published:** 2025-12-29

**Authors:** Aala A. Abulfaraj, Lina Baz

**Affiliations:** 1Biological Sciences Department, College of Science & Arts, King Abdulaziz University, Rabigh 21911, Saudi Arabia; 2Department of Biochemistry, Faculty of Science, King Abdulaziz University, Jeddah 21589, Saudi Arabia; lbaz@kau.edu.sa

**Keywords:** RNA-Seq, redox balance, antioxidant defense, osmotic adjustment, carbon flux, membrane stabilization, energy reprogramming, KEGG pathways

## Abstract

This study investigates the adaptive mechanisms that enable a single wild barley (*Hordeum spontaneum*) accession to withstand extreme salinity. Salt stress reshapes plant metabolism and gene expression, offering targets for breeding salt-tolerant cereals. A time-course RNA-Seq experiment was conducted on leaves exposed to 500 mM NaCl, followed by differential expression and functional annotations to characterize transcriptomic responses. Transcriptomic profiling identified 140 dynamically upregulated genes distributed across 19 interconnected metabolic pathways, with phased activation of oxidative phosphorylation, nitrogen assimilation, lipid remodeling, and glutathione metabolism. Central metabolic nodes, including acetyl-CoA, hexadecanoyl-CoA, and ubiquinone, coordinated bioenergetic output, membrane stabilization, and redox homeostasis. Ribose-5-phosphate and ribulose-5-phosphate linked glycolysis and the pentose phosphate pathway, supplying NADPH for antioxidant defense and nucleotide repair, while riboflavin derived from Ru5P enhanced flavoprotein activity. In parallel, glucose and fructose-6-phosphate supported osmotic adjustment and glycolytic flux, and increased sterol and cuticular lipid biosynthesis, including cholesterol-like compounds, reinforced membrane integrity and calcium signaling. Glutathione and N-acetyl-glutamate together mitigated oxidative stress and modulated polyamine metabolism, strengthening cellular resilience under salt stress. These findings outline a coordinated network of metabolic and redox pathways that can guide the engineering of salt-tolerant cereals for sustainable production in saline agroecosystems.

## 1. Introduction

Salt stress is a major abiotic threat to global food security, affecting cereal crop productivity across diverse agroecosystems. Wild barley, particularly *Hordeum spontaneum*, possesses natural salt stress tolerance mechanisms that remain incompletely characterized at the molecular level. This study focuses on a single wild barley (*H. spontaneum*) accession, for which a 500 mM NaCl treatment was employed as an extreme salinity regime, chosen on the basis of prior evidence indicating that such a high salt concentration imposes severe stress capable of amplifying salt-responsive transcriptional signatures and revealing tolerance mechanisms that are largely absent in cultivated barley, which typically withstands only comparatively mild salinity levels [1,2,3,4,5]. Wild barley (*Hordeum vulgare* sbsp. *spontaneum*), the ancestral form of cultivated barley (*H. vulgare*), is a diploid species with 2n = 2x = 14 chromosomes and a genome size that varies from about 5.7 to 9.4 pg/2C, largely due to retrotransposon-driven genomic expansion [6,7]. Native to the Fertile Crescent, this species extends its natural range into North Africa, including Egypt, where it successfully inhabits diverse and demanding environments characterized by high salinity and aridity [6,8,9]. Morphologically, *H. spontaneum* is distinguished by its brittle rachis and hulled grains, and it is renowned for its remarkable adaptability to abiotic stresses, a trait underpinned by its substantial genetic diversity [8,10].

Molecular and transcriptomic analyses, including studies on Egyptian germplasm, have demonstrated that salt stress tolerance in *H. spontaneum* involves the upregulation of ion transporters, as well as the signal transduction cascades [3,8,10,11]. Genes associated with osmoprotectant biosynthesis, including those encoding alanine/glyoxylate aminotransferase and asparagine synthetase, are also induced in cereals under salinity, contributing to cellular homeostasis [12]. However, these observations remain largely fragmented: how do energy metabolism, carbohydrate signaling, redox balance, and lipid remodeling orchestrate temporally across a salt stress time course? Which metabolites serve as central hubs linking multiple adaptive pathways? How do the dynamics of gene expression translate into measurable shifts in metabolic capacity? [13,14]. Moreover, the identity of key metabolic nodes that orchestrate cross-pathway communication under salinity stress is not well defined. This gap is particularly acute for wild crop relatives, where transcriptome-to-phenotype linkages could reveal novel genetic targets for crop improvement [15,16]. While previous studies have documented general stress responses, we still lack a time-resolved systems-level view of how wild barley coordinates energy metabolism, ion regulation, redox balance, and lipid remodeling under lethal salinity [17,18].

This work applies RNA-Seq transcriptomics combined with enriched KEGG pathway enzymes in wild barley (*Hordeum spontaneum*) to map the temporal dynamics of gene expression and metabolic rewiring in wild barley leaves under acute salt stress. By tracking gene expression across a controlled time course (0 h, 2 h, 12 h, 24 h) and linking enzymatic changes to metabolic pathways, we aim to identify: (1) the temporal sequence of metabolic pathway activation; (2) central metabolites that coordinate multiple adaptive responses; and (3) conserved, targetable mechanisms suitable for engineering salt resilience in cultivated cereals. This study infers pathway-level metabolic regulation from transcriptomic data rather than direct metabolite measurements. The novelty of this work resides in its application of an acute salinity challenge over short-term (0–24 h) RNA-Seq time points in an Egyptian wild barley ecotype, coupled with integrated pathway analysis to reveal how central metabolic hubs, rather than individual genes, jointly regulate energy metabolism, ion homeostasis, redox status, and membrane remodeling under lethal salinity.

## 2. Results

### 2.1. Quality Assessment, and Multi-Dimensional Validation of Salt-Stressed Wild Barley (Hordeum spontaneum) RNA-Seq Dataset

The RNA-Seq alignment statistics in Appendix A show that all wild barley leaf samples subjected to salt stress were sequenced at high depth (approximately 25–28 million reads per library) and mapped to the reference genome with exceptionally high overall alignment rates, consistently above 99.2%. The fraction of unmapped reads remained very low (about 0.69–1.21%), while the vast majority of reads aligned either uniquely or to multiple loci, with multi-mapping reads predominating as expected in complex, isoform-rich transcriptomes. The close similarity in alignment metrics across time points and biological replicates indicates highly reproducible library preparation and sequencing performance, providing a robust foundation for downstream transcriptomic and differential expression analyses.

The raw RNA-Seq dataset underwent further stringent quality evaluation using FastQC [19]. As illustrated in Appendix A, the per-base sequence quality distribution revealed high data fidelity, with median Phred scores consistently exceeding 30 (Q30) across the complete 90 bp read span in all 22 paired-end libraries, corresponding to a base-calling accuracy surpassing 99.9%. The per-sequence mean quality plot (Appendix A) exhibited a pronounced unimodal distribution centered around Phred 37–38, signifying that the overwhelming majority of individual reads possessed excellent overall quality, with only a negligible proportion falling within the lower-quality intervals (orange/red regions, Phred < 28). These indicators collectively affirm the superior integrity of the unprocessed sequencing data prior to alignment.

Post adapter excision and quality trimming via Trimmomatic (v0.39), the refined read sets preserved this elevated quality standard (Appendix A). The analysis revealed minimal sequence overrepresentation, with most samples displaying less than 1% of total reads flagged. Such marginal enrichment is characteristic of high-quality RNA-Seq libraries and typically arises from minor technical artifacts—such as residual adapter fragments, low-complexity poly-A/T tracts, or excessively abundant endogenous transcripts (e.g., rRNA or constitutively expressed housekeeping genes). Notably, no sample exhibited aberrantly elevated or atypical levels of overrepresented sequences, underscoring the absence of substantial contamination, preparation bias, or PCR amplification skew. Although a subset of samples (particularly Bs2_24 and Bs3_12) demonstrated slightly higher proportions relative to others, these remained well within acceptable analytical thresholds and are unlikely to exert any detrimental effect on subsequent alignment or quantitative expression profiling. As depicted in Appendix A, the bar plots delineate unique versus duplicate read fractions. All dataset contained a substantial proportion of duplicates—expected in high-depth transcriptomic libraries dominated by strongly expressed transcripts—while still retaining ample unique reads (~6–8 million per file) to ensure exhaustive transcriptome coverage and statistically robust differential expression analyses. Overall, these metrics conclusively verify that the raw RNA-Seq data derived from salt-stressed wild barley leaves possess high structural integrity and negligible sequence redundancy, thus justifying progression to subsequent read trimming, alignment, and gene expression quantification procedures.

Appendix A summarizes the distribution of reconstructed protein-coding genes across successive percentage-coverage bins (Pcov) based on their alignment to the reference protein database, together with the cumulative counts calculated from the highest toward lower coverage classes. There are 5051 genes with 100% coverage, and when adding those with 90–<100% coverage (1747 genes), the cumulative total reaches 6798 genes covered at ≥90%. Including 80–<90% coverage adds 1106 genes (7904 ≥ 80%), and continuing through decreasing bins (down to 10–<20%) yields a final total of 11,695 reconstructed genes represented in the assembly.

Then, we have aligned the high-quality RNA-Seq dataset to the wild barley accession OUH602 (*Hordeum spontaneum*) reference genome to facilitate the elucidation of novel functional mechanisms driving salt stress adaptation, complementing previously documented pathways. The results in Appendix A, and Appendix A refer to cluster analysis performed on leaf transcriptome dataset of 15-day-old wild barley (*H. spontaneum*) seedlings exposed to salt stress (500 mM NaCl) at four-time intervals (0 h, 2 h, 12 h, and 24 h). This dataset was validated consulting four distinctive approaches. They are PCA (principal component analysis), two-dimensional heatmap (Sample-to-sample correlation matrix), MA (Mean-Average) plotting (Figure 1, Appendix A, respectively), and qPCR (Appendix A). These four approaches synergistically validated the dataset, with PCA confirming global transcriptional trends, the heatmap ensuring technical precision, MA plots reinforcing biological logic, collectively providing a robust, multi-dimensional verification framework, and qPCR validating the RNA-Seq dataset by corroborating the distinct expression profiles of the analyzed transcripts across different temporal windows.

The results in PCA plot (Figure 1) reveal distinct clustering of samples according to their respective time points, indicating clear temporal separation in gene expression responses to salt stress. Furthermore, the figure demonstrates strong concordance among biological replicates at each time interval, underscoring the reproducibility and consistency of the transcriptomic data collected across the salt stress time course. Appendix A presents the two-dimensional heatmap for RNA-seq dataset derived from barley leaves subjected to salt stress. The color gradient ranges from purple (negative correlation, −0.5) to yellow (high positive correlation, 1), with black indicating no correlation (0). Samples from the same time interval were shown to cluster together, exhibiting strong intra-group positive correlations (yellow blocks along the diagonal), validating the reproducibility and consistency of biological replicates. In contrast, inter-group comparisons—particularly between 0 h and later time points (2 h, 12 h)—show reduced or negative correlations (purple blocks), reflecting substantial transcriptomic divergence induced by salt stress over time.

While Appendix A displays MA plots where each MA plot visualizes the relationship between the log2 fold change (M) and the average expression (A) for all detected transcripts when comparing different time points. In these plots, most genes cluster around a log2 fold change of zero, indicating stable expression, while differentially expressed genes are represented as outliers above or below this central cloud. The MA plots reveal a clear spread of upregulated and downregulated transcripts at each salt stress interval, with distinct patterns emerging between early (2 h, 12 h) and prolonged (24 h) exposures. This visualization confirms the presence of significant transcriptional changes in response to salt stress and validates the RNA-seq dataset by demonstrating both the dynamic range and the reproducibility of gene expression measurements across conditions. Transcripts in the latter time point shows tendency to perform like those at 0 h indicating that most regulated genes and mechanisms require 12 h to give the required level and time of expression. The qPCR measurements presented in Appendix A exhibited excellent concordance with the RNA-Seq-inferred expression trajectories reported in Appendix A for the representative genes spanning the 10 expression groups (a–j), thereby constituting a stringent, orthogonal quality-assurance tier that substantiates the robustness of the overall transcriptomic profiling.

### 2.2. Temporal Dynamics of Gene Ontology (GO) Category Enrichment in Wild Barley Under Salt Stress

Appendix A delineate the temporal distribution of transcriptomic upregulation across gene ontology (GO) categories in wild barley (*H. spontaneum*) leaves under salt stress (500 mM NaCl). Appendix A, referring to “Biological Process” category, reveals that the highest transcript levels at 2 h, 12 h, and 24 h were enriched in “response to stimulus,” “cellular process,” “biological regulation,” “regulation of biological process,” and “metabolic process,” with progressive activation of stress-responsive pathways over time. Appendix A, referring to “Molecular Function” category, highlights “catalytic activity” and “binding” as the dominant domains across all intervals, reflecting sustained enzymatic and ligand interaction roles. Appendix A, referring to “Cellular Component” category, shows enrichment in “cell,” “cell part,” “membrane,” “membrane part,” “organelle,” and “organelle part,” emphasizing membrane and organelle-associated transcript dynamics. Stacked histograms for all three figures demonstrate that these domains maintained the highest relative transcript abundances at each time point, with 2 h and 12 h exhibiting the most pronounced upregulation compared to 24 h in alignment with the results of MA (Appendix A).

### 2.3. Temporal Transcriptional Dynamics and Metabolic Enrichment of KEGG Pathway Enzymes in Wild Barley Under Salt Stress

Figure 2 presents the justifiable temporal expression patterns of transcripts (panels a–j) in 15-day-old wild barley (*Hordeum spontaneum*) leaves exposed to 500 mM NaCl at 0 h (control), 2 h, 12 h, and 24 h time points. The analysis exclusively included transcripts with “biologically justified” expression profiles: (1) up- or downregulated at a single time point (e.g., 2 h up [a], 12 h up [b], 24 h up [c], 12 h down [d], 24 h down [e]), (2) consecutively regulated across two time points (e.g., 2 h/12 h up [f], 2 h/12 h down [g], 12 h/24 h down [h]), or (3) consecutively regulated across all three stress intervals (2 h/12 h/24 h up [i], 2 h/12 h/24 h down [j]). Non-consecutive or discontinuous expression profiles (e.g., regulation at 2 h and 24 h without 12 h involvement) were excluded to prioritize temporally coherent transcriptional responses. Such intermittent expression is indicative of stochastic variability rather than the sustained, coordinated regulatory response characteristic of a genuine adaptive reaction to salt stress. Each panel illustrates distinct dynamic trends, with color-coded heatmaps or line graphs depicting fold changes relative to the control. These justifiable expression patterns were detected in 180 clusters (Table 1), out of 645 (Appendix A).

The results in Appendix A and Appendix A delineate the temporal enrichment of enzyme-encoding genes (140) across 19 interconnected KEGG pathways (Appendix A) in wild barley (*Hordeum spontaneum*) leaves under salt stress (500 mM NaCl). Enzymes exhibiting consecutive upregulation at two or three time points (2 h/12 h, 12 h/24 h, or 2 h/12 h/24 h) were prioritized, while non-consecutive profiles (e.g., 2 h/24 h) were excluded. Key enzymes include phosphoglucomutase (starch/sucrose metabolism; 2 h/12 h/24 h), ATP synthase (oxidative phosphorylation; 2 h/12 h/24 h), glutathione synthetase (glutathione metabolism; 12 h/24 h), riboflavin synthase (riboflavin metabolism; 2 h/12 h), and squalene monooxygenase (steroid biosynthesis; 12 h/24 h).

Enrichment patterns reveal progressive activation of energy production (glycolysis and the TCA cycle), ROS detoxification (glutathione metabolism), and lipid remodeling (fatty acid biosynthesis) across stress intervals. Appendix A also provide pathway-specific enzyme dynamics, including those of phosphoglycolate phosphatase (photorespiration; 2 h/12 h) and NADH dehydrogenase (oxidative phosphorylation; 12 h/24 h), underscoring adaptive metabolic rewiring.

### 2.4. Integrated Metabolic Pathways and Adaptive Mechanisms Underpinning Salt Stress Tolerance in Wild Barley

Following a comprehensive dissection of differentially enriched enzymatic components across 19 distinct metabolic and regulatory pathways, we reconstructed a network of interdependent molecular cascades that converge into discrete mechanisms of abiotic stress adaptation. This system-level interrogation uncovered six metabolic conduits (Figure 3, Figure 4, Figure 5, Figure 6, Figure 7 and Figure 8)—spanning ion homeostasis, redox balancing, osmolyte biosynthesis, energy reprogramming, membrane stabilization, etc.—that enable this wild species to circumvent ionic toxicity, osmotic dysregulation, and oxidative damage under saline duress.

Figure 3 illustrates the metabolic network remodeling in wild barley (*Hordeum spontaneum*) leaves under salt stress, emphasizing enriched enzymes within interconnected KEGG pathways centered on acetyl-CoA flux and oxidative phosphorylation. The figure identifies enzymes such as DNA helicase, DNA polymerase (archaeal type), DNA-directed RNA polymerase (RNAP), phenylalanine-tRNA ligase, signal peptidase I, peroxiredoxin, and oxaloacetate decarboxylase, which are enriched in pathways including DNA replication/repair, aminoacyl-tRNA biosynthesis, protein export, antioxidant systems, and carbohydrate metabolism. Metabolites are color-coded to distinguish metabolic trajectories: orange highlights pivotal metabolites (e.g., malonyl-CoA), pink denotes metabolites channeled toward acetyl-CoA synthesis (e.g., pyruvate), green identifies acetyl-CoA-derived intermediates, and blue marks reactions diverting to oxidative phosphorylation (e.g., TCA cycle intermediates). A red dotted arrow signifies bidirectional regulation between acetyl-CoA and hexadecanoyl-CoA, reflecting dynamic feedback in lipid and energy metabolism. The schematic underscores acetyl-CoA’s dual function as a biosynthetic precursor and metabolic nexus, alongside compensatory energy pathways, reflecting adaptive enzymatic strategies to sustain redox equilibrium and ATP production during salt exposure. Temporal enrichment of these enzymes across stress intervals highlights adaptive prioritization of energy homeostasis and stress-responsive biosynthesis. Further details on pathway-specific enzyme dynamics are provided in Appendix A.

Figure 4 delineates the temporal modulation of oxidative phosphorylation machinery in 15-day-old wild barley (*Hordeum spontaneum*) leaves exposed to salt stress, emphasizing enriched enzymatic components critical for ATP synthesis. The figure highlights upregulated subunits of mitochondrial electron transport chain (ETC) complexes—NADH dehydrogenase (Complex I), cytochrome c oxidase (Complex IV), and ATP synthase (Complex V)—which collectively drive proton motive force generation and ATP production. Temporal expression profiles demonstrate incremental activation of these complexes with prolonged salt exposure, reinforcing mitochondrial respiratory capacity to offset energy deficits. Enrichment of F-type ATPase subunits (e.g., alpha, beta, gamma) and cytochrome c oxidase assembly factors underscores adaptive prioritization of energy homeostasis under salinity. The schematic further identifies coordinated upregulation of auxiliary enzymes, including succinate dehydrogenase (Complex II) and ubiquinol-cytochrome c reductase (Complex III), ensuring efficient electron flux and redox balancing. These dynamics reflect a systemic enhancement of oxidative phosphorylation to sustain ATP-dependent processes, such as ion transport regulation and ROS detoxification, during prolonged salt stress. Appendix A provide additional granularity on enzyme-specific expression trends and pathway interactions.

Figure 5 delineates the temporal enrichment of KEGG pathway enzymes governing sugar phosphate and riboflavin biosynthesis in 15-day-old wild barley (*Hordeum spontaneum*) leaves exposed to salt stress. The figure identifies upregulated enzymes driving the synthesis of D-fructose-6P, D-ribulose-5P, D-ribose-5P, and riboflavin, with distinct color-coding demarcating three metabolic trajectories: two funneling carbon flux toward sugar phosphate intermediates (e.g., pentose phosphate pathway) and one directing riboflavin biosynthesis. A red dotted arrow denotes bidirectional regulation between D-ribulose-5P and D-ribose-5P, reflecting dynamic carbon partitioning adjustments between the pentose phosphate pathway (favoring D-ribulose-5P and NADPH generation) and ribose-5P–dependent biosynthesis (e.g., nucleotide and riboflavin synthesis), tightening metabolic control over redox balance and cofactor/nucleotide supply during salinity. Enzymatic induction—including phosphogluconate dehydrogenase (sugar phosphate synthesis) and riboflavin synthase (riboflavin production)—highlights adaptive rewiring to concurrently support osmoprotectant accumulation (via sugar phosphates) and antioxidant defense (via riboflavin-derived flavoproteins like FAD). These pathways coordinate to reconcile energy allocation with oxidative stress mitigation, ensuring redox balance and structural resilience under salinity. Appendix A provide additional details on enzyme-specific expression trends and pathway interactions.

Figure 6 delineates the temporal enrichment of glucose-centric metabolic pathways in wild barley (*Hordeum spontaneum*) leaves exposed to 500 mM NaCl. The figure identifies upregulated enzymatic components driving glucose biosynthesis through starch and sucrose metabolism and glycolysis/gluconeogenesis pathways, including phosphoglucomutase, hexokinase, and glucose-6-phosphate isomerase. These enzymes facilitate the conversion of starch-derived maltose and sucrose into glucose-6-phosphate, channeling carbon flux toward glycolysis for ATP production and osmoprotectant synthesis. Distinct color-coding highlights glucose as a central metabolite, with pathways converging to prioritize osmotic adjustment (via soluble sugar accumulation), energy replenishment (via hexose phosphorylation), and structural resilience (via cell wall polysaccharide biosynthesis). Temporal expression profiles demonstrate progressive activation of gluconeogenic enzymes (e.g., fructose-1,6-bisphosphatase) and starch-degrading enzymes (e.g., α-amylase) under prolonged salt exposure, optimizing carbon allocation to mitigate ionic and osmotic stress. The schematic underscores adaptive reprogramming of carbohydrate metabolism to sustain cellular homeostasis, with Appendix A providing further details on enzyme-specific expression trends and pathway crosstalk.

Figure 7 illustrates the temporal enrichment of enzymes of the steroid biosynthesis pathways in wild barley (*Hordeum spontaneum*) leaves exposed to 500 mM NaCl, highlighting upregulated enzymes driving the production of cholesterol and calcitriol. The figure identifies key enzymes, including squalene monooxygenase and lanosterol synthase, which are progressively activated under prolonged salt stress, channeling metabolic flux toward sterol and secosteroid biosynthesis. These pathways prioritize membrane structural stabilization (via cholesterol) and stress-responsive signaling (via calcitriol), which may modulate calcium-mediated pathways or antioxidant mechanisms. The schematic delineates cholesterol as a precursor for signaling molecules and membrane components, while calcitriol, annotated as a plant-associated sterol derivative, is implicated in redox homeostasis. Enzyme dynamics across time reflect adaptive metabolic reconfiguration to counteract osmotic and ionic stress, with Appendix A providing additional details on pathway-specific enzyme expression and interactions.

Figure 8 delineates the temporal enrichment of enzymes in glutathione-centric metabolic pathway in 15-day-old wild barley (*Hordeum spontaneum*) leaves exposed to salt stress, highlighting upregulated enzymes driving the biosynthesis of glutathione, N-acetyl-glutamate, and glycine. The figure identifies key enzymatic components, including glutamate-cysteine ligase, glutathione synthetase, and N-acetylglutamate synthase, which are progressively activated under prolonged salt stress to enhance redox homeostasis and nitrogen assimilation. These enzymes coordinate crosstalk between glutathione metabolism, amino acid biosynthesis, and nitrogen cycling pathways, with orange boxes emphasizing glutathione as a central metabolite. Glycine, a rate-limiting precursor for glutathione synthesis, is channeled through glycine cleavage systems, while N-acetyl-glutamate supports arginine biosynthesis and nitrogen flux. Temporal expression dynamics reveal incremental activation of core enzymes of these pathways, aligning with stress duration to bolster antioxidant capacity, osmotic adjustment, and detoxification. The schematic underscores possible wild barley’s reliance on glutathione-mediated ROS scavenging and metabolic flexibility to sustain cellular resilience under salinity. Further details on pathway-specific enzyme expression and interactions are provided in Appendix A.

### 2.5. Integrated GO-Based Overview of Differentially Expressed Genes

Differentially expressed genes under salt stress were distributed across core functional categories represented in all three major GO branches. In the Biological Process ontology, DEGs were mainly linked to response to stimulus, cellular process, biological regulation, regulation of biological process, and metabolic process, reflecting activation of stress perception, metabolic reprogramming, and transcriptional control particularly at 2 h and 12 h, with diminished responses at 24 h. Representative examples include genes encoding subunits of NADH dehydrogenase and cytochrome c oxidase, peroxiredoxin, and enzymes involved in glutathione-mediated redox reactions, all of which participate in redox signaling and energy-linked adaptive pathways (Figure 3, Figure 4 and Figure 8).

In the Molecular Function ontology, most DEGs encoded catalytic and binding activities—such as phosphoglucomutase, hexokinase, phosphogluconate dehydrogenase, ATP synthase and succinate dehydrogenase subunits, glutathione synthetase, riboflavin synthase, squalene monooxygenase, lanosterol synthase, various aminoacyl-tRNA ligases, DNA helicase, and DNA-directed RNA polymerase—which collectively mediate key metabolic pathways, redox balance, and macromolecular interactions. For Cellular Component, DEGs were enriched in cell/cell part, membrane/membrane part, and organelle/organelle part, indicating salt-induced remodeling of membrane-embedded and organelle-localized machineries in chloroplasts, mitochondria, and peroxisomes, including electron-transport complexes, F-type ATPase subunits, membrane-associated fatty acid and sterol biosynthesis enzymes (e.g., squalene monooxygenase, lanosterol synthase), and organellar glutathione-cycle enzymes (Figure 4, Figure 5, Figure 6 and Figure 8).

## 3. Discussion

### 3.1. Unlocking Salt Tolerance Mechanisms in H. spontaneum via OUH602 Genome-Guided Transcriptomic Profiling

In this study, we re-analyzed the raw RNA-Seq dataset from wild barley (*Hordeum spontaneum*) by aligning high-quality sequencing reads to the newly available reference genome of the wild barley accession OUH602, a resource distinguished by its chromosome-scale assembly, precise structural annotations, and comprehensive genomic coverage and historically served as a key genotype in genetic mapping, linkage analysis, and the identification of loci relevant to agronomic traits, disease resistance, and gene expression studies [20,21]. The emergence of this high-fidelity reference genome enabled us to leverage its unprecedented resolution in gene models, and syntenic relationships, providing a robust framework to dissect previously unresolved molecular mechanisms driving salt tolerance in this wild plant. By capitalizing on OUH602’s annotation accuracy and genomic completeness, we achieved enhanced mapping fidelity and identified new stress-responsive pathways and regulatory networks that underpin wild barley’s adaptive resilience to salinity.

### 3.2. Integrated Metabolic Network of Core Metabolites Underpinning Salt Stress Tolerance

The core metabolites delineated in Figure 3, Figure 4, Figure 5, Figure 6, Figure 7 and Figure 8 exhibit functional redundancy, synergistic interactions, and interdependent mechanisms, collectively underpinning multifaceted contributions to several molecular processes under saline duress. Within the metabolic network depicted in Figure 3, acetyl-CoA participates in TCA cycle, oxidative phosphorylation and lipid biosynthesis, while Hexadecanoyl-CoA and long-chain acyl-CoA esters are exclusively dedicated to structural and cuticular lipid assembly. In terms of the contribution to salt stress tolerance, acetyl-CoA plays a central role by integrating energy metabolism, and redox homeostasis [13,22,23,24,25,26,27]. In the tricarboxylic acid (TCA), acetyl-CoA fuels ATP production via oxidative phosphorylation, powering ion transporters, like *NHX1* and *SOS1*, which maintain Na^+^/K^+^ balance under salinity stress [27]. When stress inhibits conventional TCA cycle enzymes (e.g., pyruvate dehydrogenase), plants such as wheat and tomato activate alternative pathway avenues like the GABA shunt [28], which bypasses blocked steps to sustain energy flux and succinate production, critical for stress adaptation [27]. Acetyl-CoA also serves as a precursor for fatty acid and sterol biosynthesis, supporting membrane integrity through cuticular lipid synthesis [29]—a process enhanced by acyl-CoA-binding proteins (ACBPs) like ZmACBP1 in maize [30,31,32] and OsACBP4 in rice and its analog in barley [33,34,35], which stabilize lipid trafficking under salt stress [31]. Additionally, acetyl-CoA-linked metabolic rewiring in wild barley (*Hordeum spontaneum*) enhances antioxidant defenses; overexpression of acetyl-CoA-related enzymes (e.g., EkAACT in *Arabidopsis*) elevates superoxide dismutase (SOD) and peroxidase (POD) activities, mitigating accumulation of reactive oxygen species (ROS) [36,37]. These mechanisms highlight acetyl-CoA’s dual role in energy provision and stress-responsive biosynthesis. Collectively, the temporal co-induction of oxidative phosphorylation components (ETC complexes and ATP synthase) with glutathione, mercapturic acid, ubiquinone, and riboflavin-linked pathways reveals tight functional coupling between mitochondrial energy production and antioxidant defense under salinity. This coordinated activation, together with the bidirectional regulation of acetyl-CoA/hexadecanoyl-CoA and the glucose/sugar-phosphate hub, points to metabolically embedded feedback loops that redistribute carbon and reducing power to stabilize ATP supply, constrain ROS accumulation, and maintain redox homeostasis during salt stress [38,39,40]. In this context, it has been reported that acetyl-CoA, riboflavin, and their associated metabolic pathways function as a cohesive, integrated regulatory hub rather than as disparate, independent routes [27,41]. Together with sugar-phosphate–driven NADPH generation and glutathione turnover, this creates a tightly coupled network in which carbon flow, oxidative phosphorylation and riboflavin-dependent redox chemistry are co-regulated to maintain cellular energy status and redox homeostasis during salinity exposure [42,43].

Mercapturic acid plays a pivotal role in salt stress tolerance in both wild and domesticated plants by facilitating the detoxification of electrophilic and cytotoxic byproducts generated during oxidative stress, such as those arising from lipid peroxidation under high salinity conditions [44,45]. In the mercapturic acid pathway, glutathione S-transferases (GSTs) catalyze the conjugation of glutathione to reactive aldehydes and xenobiotics, forming mercapturic acid precursors that are subsequently excreted or compartmentalized, thereby preserving cellular redox homeostasis and minimizing oxidative damage to membranes and proteins [44,45]. This detoxification mechanism is integral to plant responses to salt stress, as it complements other antioxidant systems and supports the maintenance of membrane integrity and overall cellular function [46]. Enhanced glutathione metabolism and mercapturic acid production have been observed in salt-tolerant plant varieties and are considered essential for mitigating the adverse effects of ionic and osmotic stress [46,47].

Hexadecanoyl-CoA and long-chain acyl-CoA are key intermediates in the biosynthesis of cuticular waxes and membrane lipids, providing structural resilience against salt-induced water loss and ion leakage in plants [48,49,50]. In *Arabidopsis*, long-chain acyl-CoA synthetases (LACS) such as AtLACS2 activate fatty acids for cuticular lipid synthesis, and loss-of-function mutants (e.g., *atlacs2*) display increased cuticle permeability and heightened sensitivity to salt and drought, underscoring the protective role of these metabolites in stress adaptation [48,51,52]. Overexpression of *LACS* genes from apple (*MdLACS2*, *MdLACS4*) in transgenic *Arabidopsis* enhances cuticular wax accumulation and confers greater tolerance to both drought and salinity [37]. These acyl-CoAs also interact with acyl-CoA-binding proteins (ACBPs), such as OsACBP4 in rice and ZmACBP1 in maize, which facilitate lipid trafficking and help maintain membrane integrity during salt stress [24,34,48,53]. In wild barley, robust acyl-CoA metabolism supports efficient cuticle formation, while in domesticated crops, alternative splicing and regulation of *ACBP* genes further modulate lipid signaling and ion homeostasis in saline environments [53]. Collectively, the synthesis and trafficking of hexadecanoyl-CoA and long-chain acyl-CoAs underpin the structural fortification of plant surfaces and cellular membranes, representing a conserved and vital mechanism for salinity adaptation across diverse plant species [48,53].

Ubiquinone (CoQ) is essential for salt stress tolerance in both wild and domesticated plants by sustaining mitochondrial energy production and acting as a potent antioxidant [54,55]. As a key electron carrier in the mitochondrial electron transport chain, ubiquinone facilitates ATP synthesis (Figure 4), which is critical for powering ion transporters and maintaining cellular homeostasis under salinity stress [56]. Under salt-induced oxidative stress, the reduced form, ubiquinol, scavenges reactive oxygen species (ROS) and prevents lipid peroxidation, thereby protecting membrane integrity [55,57]. Transgenic tobacco plants overexpressing yeast *coq2* exhibit increased ubiquinone content, enhanced radical scavenging, and improved salt tolerance compared to wild-type plants [57]. The antioxidant function of ubiquinone is further supported by its ability to prevent DNA damage and protein oxidation, which are common consequences of salt-induced oxidative stress [57]. Thus, ubiquinone harmonizes energy metabolism and oxidative defense, underpinning plant resilience to salt stress [40].

Another sophisticated metabolic crosstalk exists among ribose-5-phosphate (R5P), ribulose-5-phosphate (Ru5P), riboflavin, and fructose-6-phosphate (Fru-6P), orchestrated through the pentose phosphate pathway (PPP) and glycolytic flux (Figure 5). Enhanced PPP activity under salt stress increases the flux of Fru-6P toward Ru5P and subsequently R5P, prioritizing the production of riboflavin and nucleotides [13,58]. In terms of the contribution of these four metabolites to salt stress tolerance, riboflavin (vitamin B_2_) bolster antioxidant defenses and maintain ionic homeostasis under the stress [59,60]. As a precursor to flavin coenzymes (FMN/FAD), riboflavin supports redox metabolism and mitigates reactive oxygen species (ROS) accumulation under salinity [43,61,62]. Exogenous riboflavin application in rice (*Oryza sativa*) reduces lipid peroxidation (malondialdehyde) and hydrogen peroxide levels while improving Na^+^/K^+^ balance, likely through upregulation of ROS-scavenging pathways and ion transporters [60]. In maize (*Zea mays*), seed priming with riboflavin (50–75 ppm) elevates antioxidant enzyme activities (SOD, CAT, APX) and osmolyte accumulation (proline, soluble sugars), reducing oxidative damage and improving growth under 70 mM NaCl stress [59]. Similarly, foliar riboflavin (2000 ppm) in tecoma (*Tecoma capensis*) restores photosynthetic pigments and leaf anatomy while enhancing catalase activity, countering salinity-induced growth inhibition [63]. Riboflavin’s dual function—acting as a non-enzymatic antioxidant and modulating enzymatic antioxidant systems—alongside osmotic adjustment, underscores its roles in plant salinity resilience [59,60,63].

Ribulose-5-phosphate (Ru5P) and ribose-5-phosphate (R5P) are pivotal intermediates in the pentose phosphate pathway (PPP), each playing distinct but interconnected roles in riboflavin biosynthesis and downstream salt stress adaptation in plants [64,65]. Ru5P acts as a direct precursor for riboflavin that supports redox homeostasis and antioxidant responses under salinity stress [13]. Enhanced PPP activity under salt stress increases NADPH production, fueling antioxidant systems and promoting Ru5P availability for riboflavin synthesis [58]. R5P, generated from Ru5P by ribose-5-phosphate isomerase, is crucial for nucleotide biosynthesis, supporting DNA repair and stress-responsive RNA synthesis, which are vital for cellular recovery under salt stress [58,66]. Notably, studies have shown that salt-tolerant genotypes exhibit increased R5P level, while salt-sensitive ones show a decline, indicating a metabolic shift that prioritizes both nucleotide and riboflavin biosynthesis for stress adaptation [66]. Additionally, the balance between Ru5P and R5P flux (Figure 5) influences the allocation of resources between riboflavin-dependent redox metabolism and nucleic acid synthesis, integrating energy, antioxidant defense, and repair mechanisms during salt stress [13,58,66].

Fructose-6-phosphate (Fru-6P) plays a central role in plant salt stress tolerance by acting as a metabolic hub that links glycolysis, the Calvin cycle, and the oxidative pentose phosphate pathway (OPPP) [27,67,68,69]. Under salinity, Fru-6P serves as a precursor for sucrose and starch biosynthesis, contributing to osmotic adjustment through the accumulation of compatible solutes that help maintain cell turgor and ion balance [27,68,70,71]. Salt stress can alter the activity of key enzymes such as fructose-2,6-bisphosphatase and pyrophosphate:fructose-6-phosphate 1-phosphotransferase (PFP), which regulate Fru-6P levels and its partitioning between glycolysis and sucrose synthesis, thereby influencing energy production and sugar homeostasis [67,69]. Plants often upregulate enzymes like fructose-1,6-bisphosphatase and fructose-1,6-bisphosphate aldolase to sustain Fru-6P flux and support continued carbon metabolism under stress [13,27]. Additionally, Fru-6P can be converted to glucose-6-phosphate, feeding into the OPPP to produce NADPH, which is essential for antioxidant defense and redox homeostasis under salt-induced oxidative stress [68]. These integrated functions of Fru-6P in sugar interconversion, energy metabolism, and antioxidant support are fundamental for plant adaptation and resilience to salinity. In conclusion, the four metabolites—R5P, Ru5P, riboflavin, and Fru-6P—are metabolically interconnected and collectively contribute to salt stress tolerance in plants by supporting energy metabolism, redox balance, and biosynthetic capacity.

Glucose has been empirically validated as an indispensable metabolic nexus for salt stress tolerance, synthesized through the enzymatic hydrolysis of di- and polysaccharides orchestrated by a suite of upregulated hydrolytic enzymes (e.g., amylases, invertases) and catabolic pathways (e.g., starch degradation, sucrose mobilization) (Figure 6 and Appendix A). Glucose contributes significantly to salt stress tolerance in plants by acting as an osmoprotectant, an energy source, and a signaling molecule [14,43,46,72,73,74]. Under salinity, glucose levels often increase in mature tissues, aiding osmotic adjustment and helping cells retain water and maintain turgor, while also supporting the accumulation of other compatible solutes like proline and sucrose [72,75]. Additionally, glucose and its derivatives can modulate stress-responsive gene expression and hormone signaling pathways, further enhancing the plant’s adaptive capacity [76,77]. However, the response is tissue- and time-dependent, with glucose sometimes declining in younger tissues or under prolonged severe stress, reflecting dynamic metabolic adjustments [72]. Overall, glucose’s roles in energy provision, osmotic balance, and stress signaling make it a central player in plant salinity resilience.

Cholesterol and calcitriol are two enriched metabolites (Figure 7 and Appendix A and Appendix A) intricately involved in plant responses to salt stress through their effects on membrane structure, signaling, and ion homeostasis. Cholesterol, as a principal phytosterol, fortifies membrane rigidity and reduces permeability, thereby limiting Na^+^ influx and helping maintain osmotic balance under saline conditions [51,78,79]. This structural reinforcement is a recognized component of plant salt stress adaptation, as sterols are known to modulate membrane integrity and stress signaling pathways [80]. Although calcitriol (1,25-dihydroxyvitamin D_3_) is primarily characterized in animal systems, its functional analogs in plants are hypothesized to influence salt stress adaptation through modulation of calcium signaling, which is tightly linked to MAPK (mitogen-activated protein kinase) cascades [81]. Under salt stress, calcium influx serves as a key second messenger, activating enzymes of the MAPK pathways that orchestrate hormonal balance, antioxidant responses, and ion homeostasis [81,82,83]. The interplay between cholesterol-mediated membrane stabilization and calcitriol-related regulatory pathways thus exemplifies a multifaceted metabolic and signaling axis, enhancing plant resilience to ionic and osmotic stress through both structural and transcriptional reprogramming [24,78,79,80,84].

Glutathione (GSH), glycine, and N-acetyl-glutamate (NAG) each play interconnected and complementary roles in plant salt stress tolerance (Figure 8). Glutathione, a key antioxidant, is known to mitigate oxidative damage by scavenging reactive oxygen species (ROS) via the AsA-GSH cycle and regenerating reduced GSH from its oxidized form (GSSG) through glutathione reductase (GR) activity [85,86,87]. It also detoxifies cytotoxic methylglyoxal via the glyoxalase system, a critical mechanism in salt-tolerant species like rice (*Oryza sativa*) [88,89,90]. Glycine, as a precursor for glycine betaine (GB), acts as an osmoprotectant, stabilizing membranes and proteins while improving Na^+^/K^+^ homeostasis. Exogenous GB in maize (*Zea mays*) enhances plasma membrane H^+^-ATPase activity, promoting Na^+^ efflux and upregulating ion transporter genes (*ZmNHX1*), thereby reducing ionic toxicity [91,92]. NAG, a precursor in arginine biosynthesis, influences polyamine metabolism and nitric oxide (NO) signaling. Overexpression of N-acetylglutamate kinase (NAGK) in plants enhances drought and salt tolerance by modulating arginine-derived polyamines (e.g., spermidine, spermine), which regulate osmotic balance and stress-responsive gene expression [93]. Notably, glutathione interacts with polyamine metabolism—exogenous GSH suppresses polyamine synthesis enzymes (e.g., arginine decarboxylase) while enhancing detoxification pathways, highlighting cross-talk between redox and osmotic responses [90,93]. Overall, the metabolites glutathione (GSH), glycine, and N-acetyl-glutamate (NAG) synergistically enhance plant salt stress tolerance through redox regulation, osmotic adjustment, and ion homeostasis. Together, these metabolites form an integrated defense network: glutathione counters oxidative stress, glycine-derived GB stabilizes ion and water balance, and N-acetyl-glutamate supports signaling and osmotic adjustment, collectively fortifying plants against salinity.

In summary, wild barley exhibited a concerted upregulation of oxidative phosphorylation, glycolysis/TCA, glutathione metabolism, nitrogen assimilation, and lipid/sterol remodeling, with central roles for acetyl-CoA, sugar phosphates, glutathione, and sterols in supporting energy supply, redox balance, osmotic adjustment, and membrane stabilization under salinity. Similar pathway-level responses have been reported in wheat and triticale, where RNA-Seq analyses under salt stress identified strong enrichment of oxidative phosphorylation, glutathione metabolism, starch and sucrose metabolism, and oxidation–reduction processes, along with upregulation of ROS-scavenging systems and sugar metabolism genes that contribute to salt tolerance. In rice, combined transcriptomic and lipidomic studies have shown that salt stress triggers changes in membrane lipid composition and activates glycerophospholipid and sphingolipid metabolism, together with glutathione-linked antioxidant pathways, to preserve membrane integrity and mitigate oxidative damage, which echoes the lipid remodeling and redox modules highlighted in wild barley [94,95,96,97,98]. Notably, the central role we attribute to acetyl-CoA, glutathione, and riboflavin in coordinating energy supply, redox buffering, and membrane stability under salinity is broadly consistent with observations in other salt-tolerant cereals such as rice and wheat. In rice, salt-tolerant genotypes show reinforced TCA cycle and oxidative phosphorylation, glutathione-dependent antioxidant systems, and changes in flavin/cofactor metabolism under salinity [94,99]. In wheat and triticale, RNA-Seq and metabolomic studies similarly report upregulation of energy metabolism, glutathione and ascorbate–glutathione cycle components, and ROS-scavenging enzymes as key features of salt tolerance Balasubramaniam [43,96,98]. Comparative analyses between wild and cultivated barley and other cereals indicate that wild germplasm typically activates these conserved modules earlier and more strongly, leading to superior maintenance of growth and water status under salt stress [3,4,36,100].

Given that the functional roles attributed to metabolites such as cholesterol, calcitriol, mercapturic acid, riboflavin, glycine, and sugar phosphates in this study are inferred from KEGG-based enzyme enrichment data rather than direct quantification, we strongly recommend that future investigations utilize metabolomic methodologies to definitively confirm their presence, characterize their temporal dynamics under salt stress, and rigorously validate their hypothesized contributions to salt tolerance in wild barley.

## 4. Materials and Methods

The raw RNA-Seq dataset for leaves of 14-day-old wild barley (*H. spontaneum*) seedlings subjected to salt stress (500 mM NaCl in half-strength Hoagland solution) at 0 h, 2 h, 12 h, and 24 h intervals were retrieved from NCBI BioProject PRJNA227211 [101]. Seeds were collected from a natural habitat in Rafah, North Sinai, Egypt (31.313559 N, 34.205973 E), located near the Mediterranean coast (salinity ∼38 g/L). The seeds were germinated and grown in a greenhouse for two weeks in trays containing a 1:1 mixture of vermiculite and perlite, maintained at 22 °C with 80% relative humidity and a 14 h photoperiod, and irrigated with half-strength Hoagland solution. Subsequently, 15-day-old seedlings were subjected to salt stress by treatment with 500 mM NaCl (29.22 g/L) dissolved in the nutrient solution for durations of 0 h, 2 h, 12 h, and 24 h. Leaf samples were harvested from individual plants in triplicate for each stress interval, with the exception of the 0 h control, for which two replicates were collected. All harvested tissues were immediately flash-frozen in liquid nitrogen and stored at −80 °C until analysis.

Total RNA isolation was conducted on morphologically homogeneous emergent flag leaf specimens harvested from three independent biological replicates at each of the four designated temporal intervals utilizing the Trizol reagent-based isolation protocol^®^ (Invitrogen, Life Tech, Grand Island, NY, USA), followed by DNase treatment (Promega Corporation, Madison, WI, USA) with 1 U/µL RNasin^®^ Plus RNase Inhibitor (Promega) at 37 °C for 2 h. Quantified RNA (30 µg at 400 ng/µL) underwent deep sequencing at BGI-Shenzhen, using the Illumina MiSeq platform, yielding ≥ 100 million raw reads per sample. Sequencing paired-end libraries were constructed at BGI-Shenzhen using the TruSeq RNA Sample Preparation kit (Illumina, San Diego, CA, USA) according to the manufacturer’s instructions [102,103]. The raw sequencing data were generated in FASTQ format and subsequently deposited in the gene bank. The quality of the raw RNA-Seq data was rigorously assessed for all libraries using FastQC (v0.11.9) [19] to evaluate per-base sequence quality and per-sequence quality score distributions. Then, the adapter-contaminated and low-quality bases were trimmed using Trimmomatic (v0.39; sliding window: 4 bp, minimum quality: 20), with read integrity verified pre- and post-processing [21,104]. Then, high-quality clean reads (≥2 mismatches) were mapped to the reference OUH602 wild barley genome using HISAT2 (v2.2.1), and Bowtie aligner (Bowtie v0.12.1) and the resulting alignment rates and specificity (unique vs. multi-mapping reads) were calculated and plotted for all biological replicates [105,106]. The reference barley genome was selected for its relevance and comprehensive characterization in barley genetic and transcriptomic studies in addition to the resulting structural annotation accuracy (https://github.com/BU-ISCIII/rnaseq-nf, 1 November 2025) [20,21]. Genome-guided de novo transcriptome assembly was conducted using Trinity RNA-Seq (v2.14.0), and its quality was validated against established benchmarks [105,106,107], with assembly performance specifically assessed by examining the completeness of reconstructed protein-coding genes through alignment of the assembled transcripts to a reference protein database. Transcript abundance was quantified with RSEM (v1.3.3), which estimates isoform-level expression values. Differential expression analysis was conducted using edgeR (v3.40.2) with a significance threshold of |log2(fold change)| ≥ 1 and false discovery rate (FDR) < 0.05 [20]. edgeR estimates dispersion using all available samples and fits negative binomial generalized linear models (GLMs) that do not require equal replication numbers across groups [108]. Expression profiles lacking temporal continuity, such as non-consecutive regulation at 2 h and 24 h, were excluded. Differentially expressed transcripts were annotated using Trinotate (v3.2.2). For validating the bioinformatics dataset, significant Pearson correlation was determined during permutation analysis and principal component analysis (PCA) was determined using trinity (v2.3.2, PtR module) with default parameters. KEGG Orthology (KO) terms were assigned via the KofamScan package (v1.3.0) with an E-value cutoff of 1e^−5^. Gene Ontology (GO) terms were analyzed using DAVID (v6.8) with default settings.

qPCR was executed, following the protocol of Bahieldin and colleagues [101], utilizing RNA aliquots generously provided by the original authors upon request. The test was used as an orthogonal validation and quality-assurance tier, wherein transcripts from representative wild barley genes spanning the 10 expression groups (a–j) depicted in Figure 2 were quantitatively profiled to substantiate the RNA-Seq–inferred transcriptional landscapes. Primers were designed with Netprimer (http://www.premierbiosoft.com/netprimer/index.html, 1 November 2025) as detailed in Appendix A, using *actin* and *glyceraldehyde-3-phosphate dehydrogenase* (*GAPDH*) as dual reference housekeeping genes to normalize qPCR measurements. For each of the 10 defined expression clusters, the second biological replicate (R2) was designated as the canonical sample for its respective time point (0, 2, 12, and 24 h), thereby serving as the representative transcriptional profile for that condition across the validation analyses.

Enrichment analysis of KEGG pathway enzymes was conducted via KEGG Mapper, prioritizing pathways with Bonferroni-corrected *p*-values < 0.01 [20]. Transcripts with logic differential expression across time in the different pathways were selected for further investigations. As several of the encoding genes participate in more than one pathway, we have studied crosstalking across pathways and generated specific functional routes to detect the core metabolites under salt stress and the possible contribution of core enriched metabolites to salt stress responses. KEGG pathway enrichment and mapping analyses were performed to identify key pathway bottlenecks and central metabolic nodes associated with transcriptomic responses to salt stress. It is also worth mentioning that the functional roles attributed to metabolites are inferred from pathway annotations (e.g., KEGG and related biosynthetic pathway information).

In general, the present work has several inherent limitations, including the use of a single wild barley accession, and the lack of accompanying physiological and metabolomic validation. Future studies involving multiple genotypes, more agronomically relevant salt regimes, and integrated physiological and metabolite profiling will be required to substantiate and extend the proposed pathway-level inferences.

## 5. Conclusions

In conclusion, this study uncovers a coordinated transcriptional program that enables wild barley (*Hordeum spontaneum*) to tolerate extreme salinity through the integration of energy metabolism, redox regulation, and osmotic adjustment. Salt-induced remodeling of oxidative phosphorylation, nitrogen assimilation, lipid metabolism, and glutathione-dependent pathways converged on central metabolic nodes that stabilized cellular membranes, sustained ATP production, and preserved redox homeostasis under 500 mM NaCl conditions. Intermediates connecting glycolysis and the pentose phosphate pathway supplied both structural carbon and reducing power, whereas enhanced sterol and cuticular lipid biosynthesis reinforced membrane integrity and supported calcium signaling. In parallel, glutathione- and N-acetyl-glutamate-linked processes mitigated oxidative damage and modulated polyamine metabolism, collectively strengthening cellular resilience to salt stress. Although these mechanisms are inferred solely from transcriptomic data and require validation by multi-omics and functional genetics, they provide a mechanistic framework and candidate targets for breeding or engineering salt-tolerant barley and other cereals suited to saline agroecosystems.

## Figures and Tables

**Figure 1 ijms-27-00358-f001:**
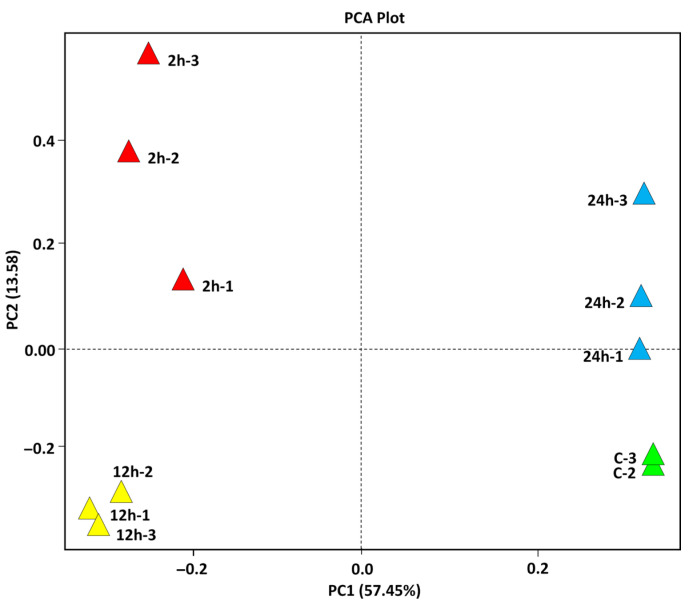
Principal component analysis (PCA) of gene expression generated from leaf transcriptome dataset of 15 d old wild barley seedlings (*Hordeum spontaneum*) exposed to salt stress (500 mM NaCl) for 0 (control, green triangles), 2 h (red triangles), 12 h (yellow triangles), and 24 h (blue triangles). Each point represents a biological replication at the indicated point. Distinct clustering of samples by treatment duration demonstrates clear temporal separation of gene expression responses to salt stress, with strong concordance among replicates at each time point.

**Figure 2 ijms-27-00358-f002:**
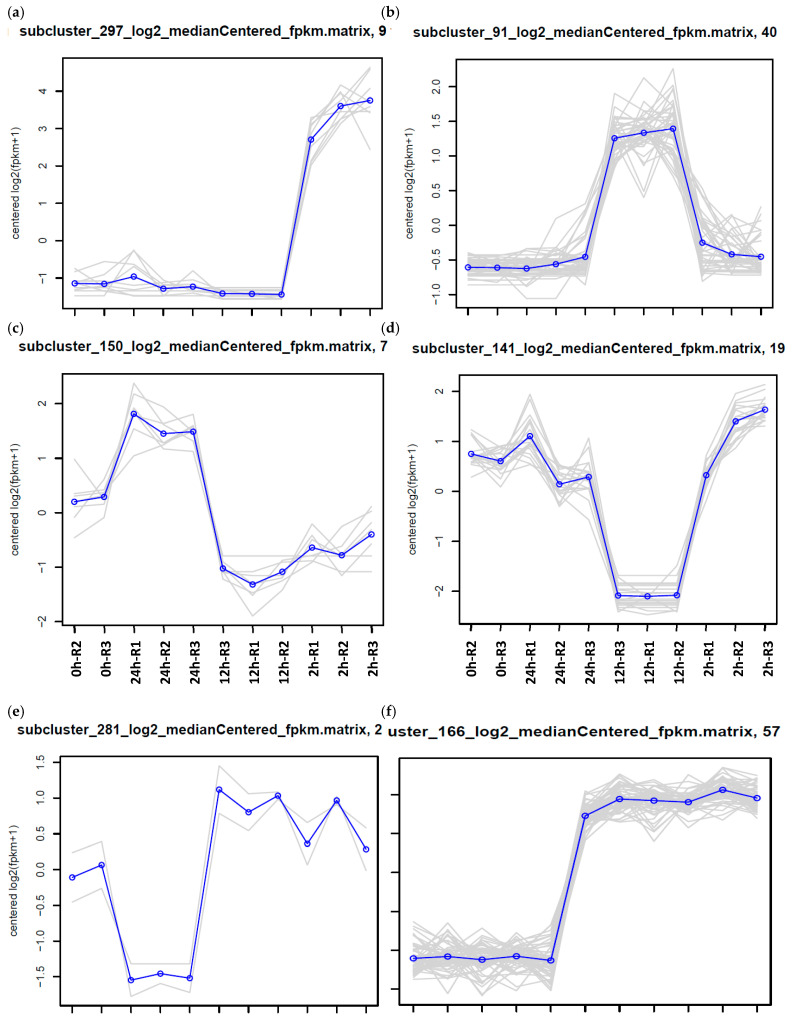
Expression patterns of transcripts (10, **a**–**j**) in 15-day-old wild barley (*Hordeum spontaneum*) leaves following salt stress treatment (500 mM NaCl) at four-time intervals, e.g., 0 h (control), 2 h, 12 h, and 24 h. Only justified expression patterns, defined as those showing up- or downregulation at a single time point, across two consecutive time points (e.g., 2 h/12 h or 12 h/24 h), or across three consecutive time points (e.g., 2 h/12 h/24 h), were included in the analysis. Expression profiles lacking temporal continuity, such as non-consecutive regulation at 2 h and 24 h, were excluded. This approach ensures that the figure highlights biologically meaningful and temporally coherent transcriptional responses to salt stress. (**a**) = 2 h up, (**b**) = 12 h up, (**c**) = 24 h up, (**d**) = 12 h down, (**e**) = 24 h down, (**f**) = 2 h/12 h up, (**g**) = 2 h/12 h down, (**h**) = 12 h/24 h down, (**i**) = 2 h/12 h/24 h up, (**j**) = 2 h/12 h/24 h down. Further information is shown in Appendix A. The grey lines denote the expression trajectories of individual transcripts within each cluster, whereas the blue line represents the mean expression trajectory of the cluster as a whole.

**Figure 3 ijms-27-00358-f003:**
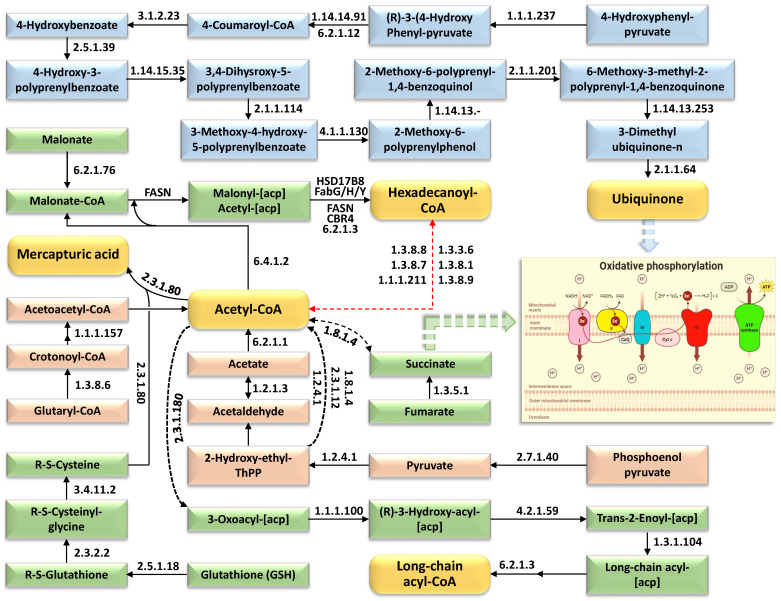
Metabolic network remodeling in wild barley (*Hordeum spontaneum*) leaves under salt stress (500 mM NaCl) across four time points (0 h, 2 h, 12 h, 24 h), illustrating crosstalk between KEGG pathways centered on acetyl-CoA flux and oxidative phosphorylation. Enriched metabolites are color-coded to distinguish metabolic trajectories: orange highlights pivotal end-products, pink denotes metabolites channeled toward acetyl-CoA synthesis (e.g., pyruvate), green identifies acetyl-CoA-derived intermediates (e.g., malonyl-CoA), and blue marks reactions diverting to oxidative phosphorylation (e.g., TCA cycle intermediates). Further information is available in Appendix A.

**Figure 4 ijms-27-00358-f004:**
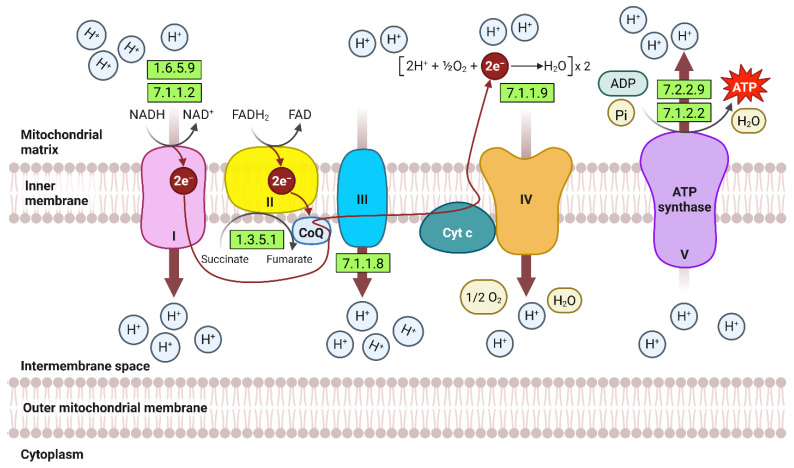
Time-dependent modulation of oxidative phosphorylation machinery in 15-day-old wild barley (*Hordeum spontaneum*) leaves under salt stress (500 mM NaCl), highlighting upregulated enzymatic components critical for ATP synthesis across four exposure intervals (0 h, 2 h, 12 h, 24 h). Further information is available in Appendix A.

**Figure 5 ijms-27-00358-f005:**
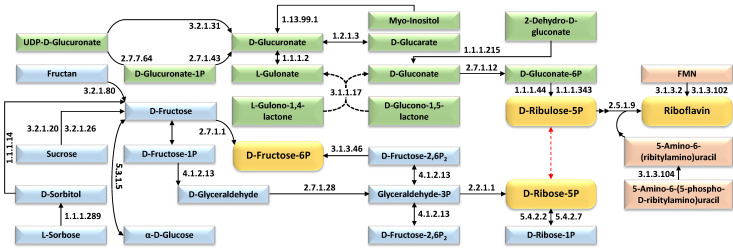
Metabolic network remodeling in salt-stressed wild barley (*Hordeum spontaneum*) 15-day-old leaves exposed to 500 mM NaCl at four intervals (0 h, 2 h, 12 h, 24 h) with temporal enrichment of KEGG pathway enzymes governing sugar phosphate and riboflavin biosynthesis under salt stress. Gene expression profiles identify upregulated enzymes driving the production of D-fructose-6P, D-ribulose-5P, D-ribose-5P, and riboflavin, illustrating cross-pathway coordination between carbohydrate metabolism and redox cofactor generation. Orange boxes highlight these pivotal end-products, while distinct color-coding delineates three metabolic trajectories, where two funneling carbon flux toward sugar phosphate intermediates and one directing synthesis of riboflavin. Further information is available in Appendix A.

**Figure 6 ijms-27-00358-f006:**
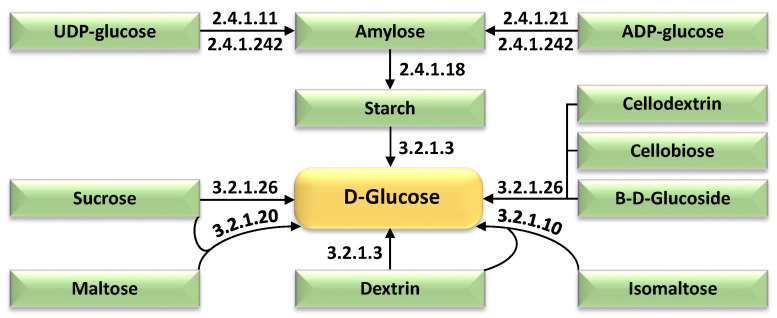
Temporal enrichment of glucose-centric pathways in 15-day-old wild barley (*Hordeum spontaneum*) leaves exposed to 500 mM NaCl across four intervals (0 h, 2 h, 12 h, 24 h). Further information is available in Appendix A.

**Figure 7 ijms-27-00358-f007:**
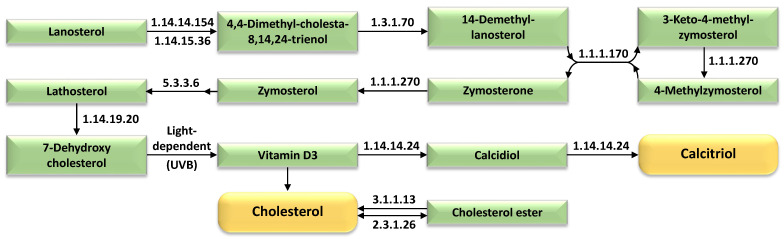
Salt stress-triggered metabolic reconfiguration in 15-day-old wild barley (*Hordeum spontaneum*) leaves subjected to 500 mM NaCl across four time points (0 h, 2 h, 12 h, 24 h), highlighting enriched enzymes of the steroid biosynthesis pathway that drives the production of cholesterol and calcitriol. Further information is available in Appendix A.

**Figure 8 ijms-27-00358-f008:**
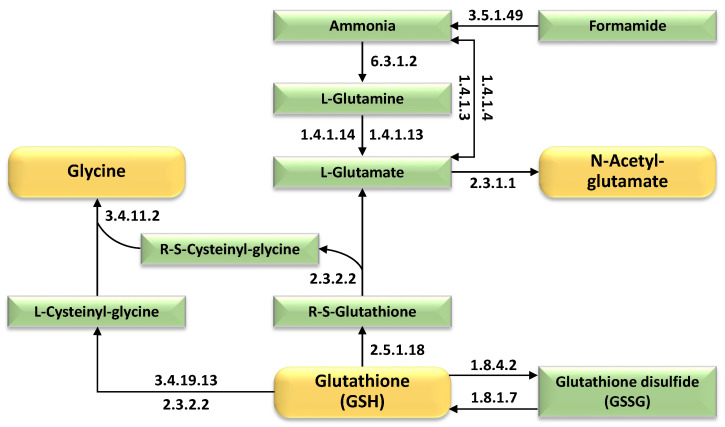
Temporal enrichment of enzymes in glutathione-centric metabolic pathway in salt-stressed wild barley (*Hordeum spontaneum*) leaves exposed to 500 mM NaCl across four intervals (0 h, 2 h, 12 h, 24 h), highlighting enzymatic upregulation driving biosynthesis of the core metabolite glutathione and critical end-products N-acetyl-glutamate and glycine. Further information is available in Appendix A.

**Table 1 ijms-27-00358-t001:** Validated expression patterns and their cluster numbers for upregulated and downregulated transcripts in 15-day-old wild barley (*Hordeum spontaneum*) leaves, analyzed across single time points (2 h, 12 h, 24 h, a–e), overlapping intervals of two consecutive time points (2 h/12 h, 12 h/24 h, f-h), or all three consecutive time points (2 h/12 h/24 h, i–j). Boldface numerical identifiers denote the transcripts selected from the 10 expression groups for validation of RNA-Seq dataset profiles via quantitative PCR.

	Expression Group	Cluster No.
a	2 h up	119, 11, 211, 230, 270, **297**, 326, 534, 600, 95
b	12 h up	129, 19, 209, 258, **284**, 478, 505, 516, 526, 61, 91, 92
c	24 h up	148, 150, 233, 290, 291, 357, 502, 509, **575**
	2 h down	NA
d	12 h down	121, 132, 136, **141**, 147, 158, 169, 176, 178, 210, 237, 253, 267, 272, 283, 286, 315, 33, 351, 355, 379, 37, 395, 427, 428, 44, 474, 488, 506, 539, 550, 56, 583, 584
e	24 h down	146, 274, **281**, 306, 327, 356, 399
f	2 h/12 h up	101, 102, 103, 106, 117, 125, 127, 128, 130, 144, 145, 151, 15, 160, 166, 180, 186, 18, 193, 201, 20, 225, 229, 238, 239, **240**, 244, 247, 24, 259, 264, 268, 269, 276, 278, 27, 311, 319, 320, 323, 34, 361, 362, 367, 374, 389, 399, 40, 429, 454, 457, 508, 541, 542, 59, 79, 88, 90, 94
g	2 h/12 h down	109, 10, 110, 118, 124, 149, **14**, 16, 172, 195, 231, 234, 235, 248, 26, 271, 285, 300, 30, 32, 372, 3, 415, 416, 446, 455, 490, 495, 511, 81, 83, 8
	12 h/24 h up	NA
h	12 h/24 h down	**204**, 467
i	2 h/12 h/24 h up	164, 222, 261, **279**, 397, 528
j	2 h/12 h/24 h down	111, 165, 194, 1, 212, 317, **377**, 387, 421, 9

## Data Availability

The original data presented in the study are openly available in the NCBI BioProject database (PRJNA227211, https://www.ncbi.nlm.nih.gov/bioproject/?term=PRJNA227211, accessed on 1 November 2025).

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
