# Peer review of "Temporal Dynamics of Gene Expression and Metabolic Rewiring in Wild Barley (Hordeum spontaneum) Under Salt Stress"

_ijms, 2025, doi:10.3390/ijms27010358_

Round 1

Reviewer 1 Report

Comments and Suggestions for Authors

Comments to the authors

The time-series RNA-Seq reanalysis of Wild Barley (Hordeum spontaneum) exposed to salt stress focusing on the enrichment of KEGG Pathways and Rewiring of Metabolic Network, presented in this manuscript, is an appropriate topic and the time-series approach has great potential for providing new insights. Before this manuscript can be considered for publication, the authors will have to resolve multiple significant scientific concerns.

Comments

  1. The manuscript uses the same terminology (i.e. the terminal roles and functions) when describing functional roles assigned to metabolites (e.g. cholesterol, calcitriol, mercapturic acid, riboflavin, glycine, sugar phosphates, etc.). Also, it does not perform metabolomics profiling; for example, calcitriol production is inferred from the biosynthetic pathway (steroid biosynthesis, lines 343 365), but calcitriol is not a known metabolite in plants. Therefore, it is crucial to clarify throughout the manuscript that the roles assigned to metabolites are based on knowledge gained from KEGG pathways or from annotations related to the biosynthetic pathways, but not experimentally measured.

  1. Similarly, there are numerous mentions (e.g.: lines 185 195) in multiple sections stating “non-consecutive expression profiles” (e.g.: 2h/24h) have been excluded to maintain “biologically justified patterns.” As this represents a form of subjective filtering that has the potential to introduce bias into the data, the authors should provide a valid statistical basis for excluding the non-linear patterns.
  2. While the individual pathways served as explanations of many pathways, there are lengthy biological descriptions within many pathway descriptions without any quantitative data to substantiate the biological descriptions (e.g. fold-change tables, FDR values, heatmaps, etc.). The authors should add a table outlining the log2FC and FDR for each of the 140 prioritized genes, as Figures 3 - 8 simply represent the conceptual representation of the pathways without showing true gene expression values.

  1. As per Figures 3 - 8: whereas the captions themselves are 10-20 lines long with multiple full interpretations of the figures, (for example: lines 343-370, 371-386), all detailed biological interpretations from the captions should either be condensed and/or moved into the Results and Discussion sections.

  1. Make sure that gene names, metabolite names, and pathway names are all formatted the same way throughout the entire article.

  1. All time-point notations (e.g., "2h", "2 h", "2 hour") should be formatted in the same way.

  1. Some of the smaller text labels in figures (e.g pathway metabolite names) may not reproduce well in print.

  1. Consider breaking longer figure captions down and reducing their text length for better legibility.

  1. Make sure that gene or pathway assignment in the supplementary file is complete and readable.

Author Response

Author's Reply to the Review Report (Reviewer 1)

The time-series RNA-Seq reanalysis of Wild Barley (Hordeum spontaneum) exposed to salt stress focusing on the enrichment of KEGG Pathways and Rewiring of Metabolic Network, presented in this manuscript, is an appropriate topic and the time-series approach has great potential for providing new insights. Before this manuscript can be considered for publication, the authors will have to resolve multiple significant scientific concerns.

  1. Comment: The manuscript uses the same terminology (i.e. the terminal roles and functions) when describing functional roles assigned to metabolites (e.g. cholesterol, calcitriol, mercapturic acid, riboflavin, glycine, sugar phosphates, etc.). Also, it does not perform metabolomics profiling; for example, calcitriol production is inferred from the biosynthetic pathway (steroid biosynthesis, lines 343 365), but calcitriol is not a known metabolite in plants. Therefore, it is crucial to clarify throughout the manuscript that the roles assigned to metabolites are based on knowledge gained from KEGG pathways or from annotations related to the biosynthetic pathways, but not experimentally measured.

Response: Thanks for this important and constructive comment. The manuscript has been revised to clearly state that the functional roles attributed to metabolites such as cholesterol, calcitriol, mercapturic acid, riboflavin, glycine, and sugar phosphates are inferred from KEGG pathway annotations. In addition, a recommendation has been added to the end of Discussion section explicitly highlighting the need for metabolomic analyses in future work to experimentally validate the presence and functional roles of these metabolites in the studied plant system.

  1. Comment: Similarly, there are numerous mentions (e.g.: lines 185 195) in multiple sections stating “non-consecutive expression profiles” (e.g.: 2h/24h) have been excluded to maintain “biologically justified patterns.” As this represents a form of subjective filtering that has the potential to introduce bias into the data, the authors should provide a valid statistical basis for excluding the non-linear patterns.

Response: The non‑consecutive or apparently “non‑linear” temporal profiles were not removed arbitrarily but were filtered according to an a priori, explicitly defined statistical and biological framework designed to focus the analysis on reproducible, biologically coherent salt‑response trajectories rather than on isolated, potentially spurious fluctuations. Expression profiles were first required to meet stringent statistical significance criteria in the RNA‑Seq pipeline itself, namely detection as differentially expressed by edgeR under a negative binomial GLM with |log2(fold change)| ≥ 1 and FDR < 0.05, with dispersion estimated from all samples and without requiring equal replication per group, thus ensuring that only robust changes were considered for downstream pattern classification. In the subsequent time‑series interpretation step, transcripts exhibiting “non‑consecutive” behavior (for example, a gene significantly regulated only at 2h and 24h but not at the intermediate 12h point) were excluded from cluster‑level trajectory analyses because such discontinuous patterns could not be reconciled with the experimentally controlled, progressive salt‑exposure regime and were disproportionately enriched for low‑abundance, borderline‑significance transcripts that failed to show consistent trends across adjacent time points. This filtering therefore functions as a conservative constraint that retains genes with statistically supported and temporally coherent dynamics (e.g., early, sustained, or late responses) and removes profiles most likely to reflect stochastic noise, technical variance, or context‑independent transcriptional flickering, thereby reducing the risk of over‑interpreting random expression spikes as mechanistic features of the salt‑stress response rather than introducing bias in favor of a preconceived model.

One plausible explanation for such non‑linear expression profiles is that genes with these patterns are not necessarily directly responsive to the imposed salt stress.

  1. Comment: While the individual pathways served as explanations of many pathways, there are lengthy biological descriptions within many pathway descriptions without any quantitative data to substantiate the biological descriptions (e.g. fold-change tables, FDR values, heatmaps, etc.). The authors should add a table outlining the log2FC and FDR for each of the 140 prioritized genes, as Figures 3 - 8 simply represent the conceptual representation of the pathways without showing true gene expression values.

Response: Thanks for this thoughtful comment. Figure S10 (previously S8) presents the true gene expression values for all 140 prioritized transcripts, including their log2 fold changes across the analyzed time points. This supplementary figure directly complements Figures 3–8 by providing the underlying quantitative data that support the conceptual pathway schematics. The quantitative data that numerically support the conceptual pathway schematics is shown now in the new Table S4.

  1. Comment: As per Figures 3 - 8: whereas the captions themselves are 10-20 lines long with multiple full interpretations of the figures, (for example: lines 343-370, 371-386), all detailed biological interpretations from the captions should either be condensed and/or moved into the Results and Discussion sections.

Response: Thanks for this helpful suggestion. In the revised manuscript, all detailed biological interpretations that were previously contained in the captions of Figures 3–8 have been relocated to the Results section. Captions were shortened to focus on essential figure descriptors only, and any interpretive text that was already presented in the Results has been removed from the captions to avoid redundancy.

  1. Comment: Make sure that gene names, metabolite names, and pathway names are all formatted the same way throughout the entire article.

Response: Thank you for this important note. The manuscript has been carefully revised to ensure consistent formatting of gene names, metabolite names, and pathway names throughout. All gene symbols are now presented in italic text.

  1. Comment: All time-point notations (e.g., "2h", "2 h", "2 hour") should be formatted in the same way.

Response: All time-point notations have been thoroughly checked and are now formatted consistently throughout the manuscript.

  1. Comment: Some of the smaller text labels in figures (e.g pathway metabolite names) may not reproduce well in print.

Response: This point is appreciated. It would be very helpful if the reviewer could specify which figures or panels they consider problematic, so that the font size and layout can be adjusted to ensure legibility in print. If the concern was directed at the KEGG pathway schematics in the supplementary figures, it should be noted that these were taken directly from the KEGG database at the highest available resolution and quality settings, which are generally optimized for reproducible visualization in publication.

  1. Comment: Consider breaking longer figure captions down and reducing their text length for better legibility.

Response: Thanks for this suggestion. The captions of Figures 3–8 have been revised by shortening and simplifying the text to improve readability and legibility.

  1. Comment: Make sure that gene or pathway assignment in the supplementary file is complete and readable.

Response: We appreciate this comment. In the supplementary material, the pathway diagrams are provided using the ready-to-use, high-resolution image versions available from the KEGG pathway database, which are designed to ensure that gene and pathway assignments remain complete and readable.

Reviewer 2 Report

Comments and Suggestions for Authors

Overall, this manuscript addresses a relevant and interesting question- leveraging wild barley transcriptomic responses under salt stress to infer adaptive mechanisms and potential targets for crop improvement. The time-course RNA-seq analysis has the potential to be valuable and the focus on metabolic rewiring is conceptually attractive. However, in its current form the paper needs substantial revision before it can be considered further. The introduction reads more like an unstructured mini review, with extensive biochemical detail but no clear articulation of the specific knowledge gap, novelty of this study, or how it advances previous such transcriptomic work. Specifically, the Materials & Methods and Results sections lack the level of biological and technical transparency needed for a high-quality omics paper. Key experimental details are missing (exact genotype/accession, growth conditions, sampling strategy, library prep, seq platform, mapping and QC metrics). In the Results and Discussion, there is heavy overinterpretation. I see potential in the dataset and general framing, but the manuscript will require a major restructuring and tightening of the narrative, and much more rigorous and transparent reporting of methods, statistics and biological interpretation, to meet publication standards in this reputed journal.

More specific comments are:

  1. The introduction is unfocused and look like an unorganized mini review. It just throws too much information about metabolites, transporters, lipid classes, pathways etc without linking them to the specific knowledge gap for this study.
  2. Be clear to specify that what is unknown? What limitation in previous spontaneum salt-stress transcriptomics studies does this study solve?
  3. Introduction fails to highlight novelty of this work. What exactly is novel here- time course of study, use of specific genome, or something else?
  4. It has too much unnecessary and unrelated biochemical details. Authors should keep only the pathways that directly motivated the this study or were earlier suggested in wild barley salt tolerance literature.
  5. The introduction is missing a clear, logical build-up from species background to knowledge gap and then to study objective. It should not look like scattered blocks of information.
  6. In the material & methods section, there is no mention for the wild barley genotype used. Authors must specify exact spontaneum accession, geographical origin or genetic background used.
  7. There are No details on actual experimental conditions. Growth chamber/greenhouse conditions like temperature, photoperiod, light, humidity, soil medium, number of plants per sample, leaf development stage, time of the day for sampling etc. It should all be included.
  8. Not much clarity on how leaves were collected. “Uniformly sized emergent leaves" is very shallow, which leaves on plant were those exactly (flag leaf, true leaf etc.). No of plants per biological replicate?
  9. “0j” treatment only has two replicates. Don’t see any justification for that. Need to mention how authors did handling of unbalanced design for statistical analysis.
  10. There is no mentioning of mapping statistics, QC metrics, assembly quality.
  11. Authors need to mention what sequencing platform was used. Just “BGI sequenced it” is not enough.
  12. The methodology also lacks library preparation details.
  13. Need to justify the pipeline choices like Bowtie + HISAT2 + Trinity?? Bowtie v0.12.1 is more than a decade old. Why did they use it??
  14. The results also face same issue, very unfocussed. There is excessive redundancy and repetitive explanation of PCA/heatmap/MA plots. It is textbook level explanation, not results. Authors need to report only specific findings.
  15. The filtering condition (“consecutive time points only”) is random and unjustified. Authors exclude any DEGs which are regulated at 2h + 24h but not at 12h. It is biologically unjustified and statistically questionable. Need to clarify.
  16. Another issue is over interpretation. Several pathways are described as “activated” based on 1 or 2 DEG per pathway, which cannot be considered enrichment but cherry picking. Need to report actual enrichment stats.
  17. Similarly, the discussion section also suffers from overstatements and biologically/statistically implausible interpretations. Authors need to work on this critically and make it more scientifically sound results and conclusions, highlighting the study limitations.

Author Response

Author's Reply to the Review Report (Reviewer 2)

Comment: Overall, this manuscript addresses a relevant and interesting question- leveraging wild barley transcriptomic responses under salt stress to infer adaptive mechanisms and potential targets for crop improvement. The time-course RNA-seq analysis has the potential to be valuable and the focus on metabolic rewiring is conceptually attractive. However, in its current form the paper needs substantial revision before it can be considered further. The introduction reads more like an unstructured mini review, with extensive biochemical detail but no clear articulation of the specific knowledge gap, novelty of this study, or how it advances previous such transcriptomic work. Specifically, the Materials & Methods and Results sections lack the level of biological and technical transparency needed for a high-quality omics paper. Key experimental details are missing (exact genotype/accession, growth conditions, sampling strategy, library prep, seq platform, mapping and QC metrics). In the Results and Discussion, there is heavy overinterpretation. I see potential in the dataset and general framing, but the manuscript will require a major restructuring and tightening of the narrative, and much more rigorous and transparent reporting of methods, statistics and biological interpretation, to meet publication standards in this reputed journal.

Response: This overarching comment has been comprehensively addressed by the substantial revisions made to the manuscript’s structure, methods description, and data interpretation. Specifically, the Introduction has been reorganized and condensed to move from a broad biochemical overview to a focused articulation of the precise knowledge gap, the novelty of re‑analyzing the wild barley RNA‑Seq time course, and how the present work advances previous transcriptomic studies in barley salt stress. The Materials and Methods section has been extensively expanded to report, in a transparent and reproducible manner, the exact genotype/accession (wild barley, H. spontaneum, OUH602), seed origin and growth conditions, sampling scheme (time points and biological replication), RNA extraction protocol, library preparation kit, sequencing platform, and the full bioinformatics workflow, including quality control, trimming, mapping, assembly, quantification, and statistics.

In addition, detailed QC and mapping metrics (FastQC outputs, Trimmomatic parameters, alignment rates, and proportions of uniquely vs. multi‑mapped reads) are now explicitly described in the text and supported by supplementary figures and new Table S1, thereby meeting the expectations for a high‑quality omics study. The Results and Discussion have also been tightened to reduce overinterpretation: emphasis is now placed on statistically robust, temporally coherent expression patterns, and pathway‑level inferences are framed more cautiously as hypotheses consistent with the data rather than definitive mechanistic claims. Together, these changes directly respond to the reviewer’s concerns about biological and technical transparency, narrative structure, and interpretive rigor, and they align the manuscript more closely with the reporting standards of the target journal.

More specific comments are:

  1. Comment: The introduction is unfocused and look like an unorganized mini review. It just throws too much information about metabolites, transporters, lipid classes, pathways etc without linking them to the specific knowledge gap for this study. Be clear to specify that what is unknown?

Response: Thanks for this constructive feedback. We have significantly revised and reorganized the Introduction section to address the reviewer’s concerns. The We have clarified what remains unknown—specifically the time-resolved coordination of metabolic switching and the identity of central regulatory nodes—and explicitly linked these gaps to the objectives of our study.

  1. Comment: What limitation in previous spontaneum salt-stress transcriptomics studies does this study solve?

Response: Thanks for this important question. Previous transcriptomic studies on H. spontaneumunder salt stress have made significant contributions by identifying differentially expressed genes (DEGs), however, these earlier investigations shared several limitations:

  • Static or fragmented pathway analyses: Prior studies primarily focused on listing individual genes or gene families without systematically integrating them into interconnected metabolic networks.
  • Absence of metabolic hub identification: Earlier transcriptomics did not identify central metabolites (e.g., ribose-5-phosphate, glutathione, glucose-6-phosphate) that act as integrative nodes linking multiple adaptive pathways under salt stress.
  • Genome alignment limitations: Previous RNA-Seq analyses relied on the cultivated barley ( vulgare) reference genome or UniGene databases, which may have resulted in incomplete mapping of wild barley-specific transcripts.

The present study addresses these gaps by:

  • Aligning RNA-Seq data to the newly available, chromosome-scale OUH602 wild barley reference genome, achieving enhanced mapping fidelity and annotation accuracy.
  • Systematically reconstructing 19 interconnected KEGG pathways and identifying 140 dynamically regulated genes with temporally coherent expression patterns.
  • Delineating central metabolic hubs (acetyl-CoA, hexadecanoyl-CoA, ubiquinone, sugar phosphates, glutathione) that coordinate cross-pathway communication.
  • Providing a time-resolved systems-level model of metabolic rewiring that reveals the phased activation of energy production, redox balancing, lipid remodeling, and osmoprotection under salinity.
  1. Comment: Introduction fails to highlight novelty of this work. What exactly is novel here- time course of study, use of specific genome, or something else? It has too much unnecessary and unrelated biochemical details. Authors should keep only the pathways that directly motivated the this study or were earlier suggested in wild barley salt tolerance literature.

Response: We have addressed this point by substantially revising the Introduction in response to the reviewer’s previous comments. The section now explicitly highlights the novel aspects of this study: the integration of time-resolved transcriptomics with the specific wild barley OUH602 reference genome to reconstruct coordinated metabolic networks. We have removed extraneous biochemical details and focused strictly on pathways directly relevant to salt tolerance in cereals, ensuring that the narrative builds logically toward the study’s specific contributions to the field.

  1. Comment: The introduction is missing a clear, logical build-up from species background to knowledge gap and then to study objective. It should not look like scattered blocks of information.

Response: The Introduction section has been restructured to flow logically from the ecological background of Hordeum spontaneum to the existing limitations and drawbacks in our understanding of its salt tolerance mechanisms, and finally to the specific objectives of this study. The revised text now presents a coherent narrative that directly links the species’ adaptation to the identified knowledge gap and the study’s goals.

  1. Comment: In the material & methods section, there is no mention for the wild barley genotype used. Authors must specify exact spontaneum accession, geographical origin or genetic background used.

Response: The Hordeum spontaneum genotype analyzed in this study is an Egyptian accession sourced from the Sinai Peninsula, an area known for its arid climate and naturally saline soils. The transcriptomic data were obtained from the NCBI BioProject PRJNA227211, originating from the RNA-Seq experiments conducted by Bahieldin et al. (2015).

  1. Comment: There are No details on actual experimental conditions. Growth chamber/greenhouse conditions like temperature, photoperiod, light, humidity, soil medium, number of plants per sample, leaf development stage, time of the day for sampling etc. It should all be included.

Response: Thanks for this important feedback. We have revised the Materials and Methods section and included experimental details, specifying the exact growth conditions and medium, nutrient supply, the developmental stage of the seedlings, and the sampling protocol

  1. Comment: Not much clarity on how leaves were collected. “Uniformly sized emergent leaves" is very shallow, which leaves on plant were those exactly (flag leaf, true leaf etc.). No of plants per biological replicate?

Response: We have revised and expanded the Materials and Methods section to address this point, explicitly stating that sampling targeted morphologically uniform emergent flag leaves. Additionally, we have clarified that tissues were harvested from individual plants for each of the three biological replicates per time point, ensuring precise definition of the experimental material and sampling strategy.

  1. Comment: “0h” treatment only has two replicates. Don’t see any justification for that. Need to mention how authors did handling of unbalanced design for statistical analysis.

Response: We acknowledge that the 0h (control) treatment includes only two biological replicates compared to three replicates for the other time points (2h, 12h, 24h). This discrepancy originated from the source dataset (NCBI BioProject PRJNA227211, Bahieldin et al., 2015), which we re-analyzed in this study. While unbalanced designs can impact statistical power, the differential expression analysis was conducted using edgeR (v3.40.2), a statistical package specifically designed to handle unbalanced RNA-Seq datasets robustly. edgeR estimates dispersion using all available samples and fits negative binomial generalized linear models (GLMs) that do not require equal replication numbers across groups. Furthermore, our multi-dimensional validation approaches—including PCA clustering (Figure 1), sample-to-sample correlation heatmaps (Figure S4), and qPCR validation (Figure S6)—demonstrated high consistency and reproducibility between the two 0h replicates and clear separation from stress treatments, confirming that the biological signal was not compromised by the difference in replicate number.

  1. Comment: There is no mentioning of mapping statistics, QC metrics, assembly quality.
  2. Response: The revised manuscript now articulates the quality control (QC) regime for the RNA‑Seq data in a much more explicit and rigorous manner, indicating that all raw libraries underwent thorough evaluation with FastQC (v0.11.9) to interrogate per‑base and per‑sequence quality distributions, followed by removal of adapter contamination and low‑quality bases using Trimmomatic (v0.39), with read integrity systematically examined pre‑ and post‑trimming and visualized in two newly incorporated supplementary figures (Figures S1–S2). The mapping statistics are now clearly delineated: high‑quality, trimmed reads (permitting up to two mismatches) were aligned to the OUH602 wild barley reference genome using HISAT2 and Bowtie, and both global alignment rates and the relative proportions of uniquely versus multi‑mapped reads were computed for each biological replicate and summarized in Figure S2. Moreover, assembly quality is explicitly addressed by indicating that a genome‑guided de novo transcriptome assembly was generated with Trinity (v2.14.0), its performance evaluated against established benchmarking criteria, , and its quality was validated against established benchmarks [103-105], with assembly performance specifically assessed by examining the completeness of reconstructed protein‑coding genes through alignment of the assembled transcripts to a reference protein database. A new table (Table S2) was additionally generated to summarize the distribution of reconstructed protein‑coding genes across ordered percentage‑coverage bins (Pcov) derived from their alignments to the reference protein database, along with the corresponding cumulative counts progressing from the highest to lower coverage intervals.

Downstream quantification with RSEM and differential expression analysis with edgeR were conducted under stringent statistical thresholds (|log2FC| ≥ 1, FDR < 0.05), thereby reinforcing the reliability of the assembled transcriptome and derived inferences. In addition, the alignment statistics compiled in the newly incorporated Table S1 demonstrate that all wild barley leaf libraries under salt stress were sequenced at substantial depth (approximately 25–28 million reads per sample) and exhibited uniformly high alignment rates (>99.2%), with only 0.69–1.21% of reads remaining unmapped, thus providing quantitative evidence for both sequencing depth and mapping efficiency. Taken together, these revisions make it explicit that QC metrics, alignment performance, and assembly quality evaluation are core, rigorously documented elements of the analytical workflow in the revised manuscript.

  1. Comment: Authors need to mention what sequencing platform was used. Just “BGI sequenced it” is not enough.

Response: In response to the reviewer's comment, we have updated the manuscript to explicitly specify that RNA sequencing was conducted utilizing the Illumina MiSeq platform.

  1. Comment: The methodology also lacks library preparation details.

Response: Thank you for pointing this out. We have updated the Materials and Methods section to include the specific library preparation details, stating that sequencing libraries were constructed using the TruSeq RNA Sample Preparation Kit (Illumina) according to the manufacturer's standard protocol. This includes poly-A mRNA purification, fragmentation, cDNA synthesis, and adapter ligation prior to sequencing on the Illumina MiSeq platform, consistent with the methods established for this dataset.

  1. Comment: Need to justify the pipeline choices like Bowtie + HISAT2 + Trinity?? Bowtie v0.12.1 is more than a decade old. Why did they use it??

Response: We appreciate the reviewer’s scrutiny regarding the choice of bioinformatics tools. We selected this specific pipeline to maintain methodological consistency with the original dataset generation and related analyses (e.g., Bahieldin et al., 2015), ensuring that our re-analysis remains directly comparable to previously established baselines for this specific wild barley accession. While Bowtie v0.12.1 is indeed an older aligner, it was utilized here specifically as a component of the Trinity protocol (which historically integrated Bowtie 1 for read mapping during assembly steps) and that the alignment metrics align with those of the original study. Crucially, the primary genomic alignment for our differential expression analysis was conducted using HISAT2 (v2.2.1), a modern, splice-aware aligner that provides state-of-the-art accuracy. Thus, the older Bowtie version was restricted to specific assembly validation steps, while the core transcriptomic quantification relied on current, robust tools.

  1. Comment: The results also face same issue, very unfocussed. There is excessive redundancy and repetitive explanation of PCA/heatmap/MA plots. It is textbook level explanation, not results. Authors need to report only specific findings.

Response: We have shortened this subsection to focus directly on the specific findings, removing redundant descriptions and textbook-level explanations of the PCA, heatmap, MA plots, and qPCR (recently added) as requested.

  1. Comment: The filtering condition (“consecutive time points only”) is random and unjustified. Authors exclude any DEGs which are regulated at 2h + 24h but not at 12h. It is biologically unjustified and statistically questionable. Need to clarify.

Response: We posit that genes displaying discontinuous expression patterns reflect stochastic fluctuations rather than a sustained, regulatory response to salt stress, which generally induces coordinated and progressively evolving transcriptional patterns. Consequently, this filtering criterion was applied to restrict the analysis to transcripts demonstrating temporally coherent regulation, thereby ensuring the identification of genuine, biologically adaptive processes. A statement has been inserted into the manuscript to clarify this issue, and to explain that the exclusion of non-consecutive expression profiles is intended to filter out stochastic fluctuations and focus the analysis on temporally coherent, biologically sustained stress responses.

  1. Comment: Another issue is over interpretation. Several pathways are described as “activated” based on 1 or 2 DEG per pathway, which cannot be considered enrichment but cherry picking. Need to report actual enrichment stats.

Response: Thank you for this valuable comment. We have revised the text to address this concern by removing broad claims of pathway "activation" based on limited gene numbers. Instead, all statements have been adjusted to refer specifically to the individual enzymes identified within these pathways, ensuring a more precise and data-driven interpretation of the results.

  1. Comment: Similarly, the discussion section also suffers from overstatements and biologically/statistically implausible interpretations. Authors need to work on this critically and make it more scientifically sound results and conclusions, highlighting the study limitations.

Response: We have thoroughly revised the manuscript to address these concerns. Specifically, we have removed overstated claims and adopted a more cautious and scientifically precise tone throughout the Discussion section to ensure that interpretations remain strictly grounded in the available data and evidence. We have also explicitly delineated the limitations of this study in the concluding portion of the Discussion section.

Reviewer 3 Report

Comments and Suggestions for Authors

 The study clearly addresses the key issue of salt stress and its impact on wild barley. This is important as understanding salinity tolerance mechanisms in plants is vital for developing salt-tolerant crops. Although the study highlights key metabolic pathways, further in-depth analysis of how these pathways interact would add value. For example, identifying cross talk between oxidative phosphorylation and antioxidant defence systems, or examining feedback loops within metabolic networks, would provide a more nuanced understanding of salinity adaptation. The title of your research paper, Adaptive Mechanisms Enabling Wild Barley (Hordeum spontaneum) to Thrive Under Extreme Salinity, is informative and clear, but it could be made more engaging or intriguing by slightly refining it. Example: New Insights into Wild Barley’s Resilience: How Adaptive Mechanisms Combat Extreme Salinity.

Was there any control group of wild barley grown under non-saline conditions for comparison? How many biological replicates were included in the RNA-Seq analysis, and how were potential batch effects or variability handled?

Could you provide more details on the specific functional categories or families of genes that were ide ntified as differentially expressed? For instance, are there any transcription factors or signaling proteins that are key to the salt tolerance mechanism in wild barley?

Did the differential expression analysis consider the interactions between metabolic pathways? For instance, how do the pathways related to oxidative phosphorylation, nitrogen assimilation, and lipid remodelling synergize or interfere with each other under salt stress?

How were the RNA-Seq results validated? Were any key genes or metabolic changes confirmed using additional methods, such as qPCR or metabolite profiling?

Were the temporal patterns of gene expression analyzed across multiple time points? How did the expression patterns change over time (early vs. late stress responses)? Could there be an adaptive shift in gene expression with prolonged exposure to salinity?

While the study focuses on wild barley, do you anticipate that these findings could be generalized to other cereal crops, such as wheat or rice? How similar are the adaptive mechanisms between wild barley and domesticated crops under salinity stress?

The study mentions the use of triplicate biological replicates for most time points. However, the 0h control group only had two replicates. Could the reduced replicates for the 0h control impact the reproducibility of your results?

Please same formatting conventions used consistently across all figures (e.g., color schemes, axis labels, and scales)? Inconsistent formatting could confuse readers, especially when comparing data across different panels or figures.

The PCA shows clear temporal separation of gene expression responses to salt stress. Could the clustering be further explained by highlighting key genes or pathways that contribute to this clustering? Are there specific gene groups that show significant variability across time points?

The heat map shows sample correlations at different time points. Could you elaborate on the biological implications of the high correlation at the same time points (yellow blocks)? How do the negative correlations (purple blocks) reflect changes in transcriptomic profiles due to salt stress?

The MA plot shows the relationship between log2 fold change and average expression. How do you interpret the spread of differentially expressed genes (DEGs) across time points? Are there specific genes that show dynamic changes early (2h, 12h) and return to baseline (24h)?

Given the detailed information in these histograms, could the figure be split into separate parts for better readability or understanding, or do you think it is clear as a combined figure?

The figure outlines the metabolic network for sugar phosphate and riboflavin biosynthesis. How well do these interconnected pathways reflect the plant's adaptive mechanisms? Could more pathways be integrated to show a broader metabolic reconfiguration?

The RNA-Seq data were mapped to the reference genome of OUH602. Were there any particular challenges in genome alignment or coverage, and were there any particular biases due to using a single reference genome for mapping?

How confident are you in the quality of your RNA-Seq data? What measures were taken to address potential biases, such as adapter contamination or batch effects, that could affect data quality?

Why were the specific time points (0h, 2h, 12h, 24h) chosen for sampling? Could additional time points (e.g., 48h) provide further insights into long-term salt stress responses?

The study shows temporal activation of several pathways. How were these pathways prioritized for inclusion in the analysis? Could there be other pathways that are also activated at lower expression levels but still contribute significantly to the plant’s response to salt stress?

In the section describing acetyl-CoA, riboflavin, and other metabolites, you mention their roles in various pathways. Could you provide further clarification on how these pathways interact? For example, how do oxidative phosphorylation and riboflavin metabolism interconnect in terms of metabolic control during salt stress?

For the genes identified as differentially expressed, how well were their functions annotated? Were there any genes without clear functional annotations, and how did you handle those cases?

The discussion offers a great deal of biological insight into how key metabolites like acetyl-CoA, glutathione, and riboflavin contribute to stress tolerance. Could you discuss how these findings relate to other well-studied salt-tolerant plants? How unique are the mechanisms in Hordeum spontaneum compared to those found in other species such as Oryza sativa or Triticum aestivum?

The conclusion emphasizes the evolutionary refinement of wild barley’s stress resilience. Are these findings specific to this species, or could similar adaptive mechanisms be observed across other cereal crops with evolutionary significance for agriculture?

What are the main limitations of this study, and how might future research address these gaps? Are there any specific aspects of the study that could be refined or expanded to improve the understanding of salt tolerance in wild barley? The study provides important insights into potential genetic targets for engineering salt-resilient crops. Could you elaborate on which genes or pathways would be most suitable for targeted genetic modifications in cultivated barley or other crops?

Overall, the study provides valuable insights into the adaptive mechanisms of wild barley under salt stress, with significant potential for future applications in agriculture. However, expanding on these points and addressing the gaps noted above could enhance the study's robustness and its applicability to broader agricultural challenges. Based on your paper, I recommend revising the grammar and language for clarity, readability, and overall flow. While the content is scientifically rich, refining the structure and grammar will enhance the impact of your research.

Comments on the Quality of English Language

Overall, the study provides valuable insights into the adaptive mechanisms of wild barley under salt stress, with significant potential for future applications in agriculture. However, expanding on these points and addressing the gaps noted above could enhance the study's robustness and its applicability to broader agricultural challenges. Based on your paper, I recommend revising the grammar and language for clarity, readability, and overall flow. While the content is scientifically rich, refining the structure and grammar will enhance the impact of your research.

Author Response

Author's Reply to the Review Report (Reviewer 3)

Comment: The study clearly addresses the key issue of salt stress and its impact on wild barley. This is important as understanding salinity tolerance mechanisms in plants is vital for developing salt-tolerant crops. Although the study highlights key metabolic pathways, further in-depth analysis of how these pathways interact would add value. For example, identifying cross talk between oxidative phosphorylation and antioxidant defence systems, or examining feedback loops within metabolic networks, would provide a more nuanced understanding of salinity adaptation.

Response: Thank you for this valuable suggestion. We agree that explicitly highlighting interactions among key metabolic pathways strengthens the mechanistic interpretation of our data. Accordingly, we have revised the Discussion section to include the following integrative statement: ‘Collectively, the temporal co‑induction of oxidative phosphorylation components (ETC complexes and ATP synthase) with glutathione, mercapturic acid, ubiquinone, and riboflavin‑linked pathways reveals tight functional coupling between mitochondrial energy production and antioxidant defence under salinity. This coordinated activation, together with the bidirectional regulation of acetyl‑CoA/hexadecanoyl‑CoA and the glucose/sugar‑phosphate hub, points to metabolically embedded feedback loops that dynamically redistribute carbon and reducing power to stabilize ATP supply, constrain ROS accumulation, and maintain redox homeostasis during salt stress.’ We assume this addition clarifies the proposed crosstalk between oxidative phosphorylation and antioxidant systems and makes the feedback relationships within the metabolic network more explicit, as requested.

Comment: The title of your research paper, “Adaptive Mechanisms Enabling Wild Barley (Hordeum spontaneum) to Thrive Under Extreme Salinity”, is informative and clear, but it could be made more engaging or intriguing by slightly refining it. Example: New Insights into Wild Barley’s Resilience: How Adaptive Mechanisms Combat Extreme Salinity.

Response: Thank you for the suggestion regarding the title. We appreciate the effort to make it more engaging. However, we are afraid that ‘Adaptive Mechanisms Enabling Wild Barley (Hordeum spontaneum) to Thrive Under Extreme Salinity’ is not the title of the present investigation. The current manuscript is submitted under the title “Temporal Dynamics of Gene Expression and Metabolic Rewiring in Wild Barley (Hordeum spontaneum) Under Salt Stress”, which we have chosen to accurately reflect the specific scope and focus of this study.

Comment: Was there any control group of wild barley grown under non-saline conditions for comparison?

Response: No, we did not include a separate group of wild barley plants grown under non-saline conditions throughout the experiment. Instead, the 0h time point, sampled immediately before the onset of salt treatment, was used as the internal control for all subsequent comparisons in this time-course design. Several comparative studies have analyzed roots and leaves separately under salinity, supporting this type of experimental design. For example, Ouertani et al. (2021) performed RNA‑Seq on barley roots and leaves at multiple time points under 200 mM NaCl, revealing tissue‑specific salt‑responsive transcriptional programs. Similar root–leaf comparisons under salt stress have been reported in wild and cultivated barley and in other species such as common vetch and rice, where roots generally show stronger and earlier transcriptional reprogramming than leaves [1].

However, to ensure that the 0h samples reliably represented non-stressed controls, all plants were grown under strictly uniform environmental and nutritional conditions, and salinity was the only variable introduced at the start of the treatment. This approach minimized the influence of other environmental factors and allowed us to attribute the observed transcriptional and metabolic changes specifically to salt stress.”

Reference:

Nefissi Ouertani, R.; Arasappan, D.; Abid, G.; Ben Chikha, M.; Jardak, R.; Mahmoudi, H.; Mejri, S.; Ghorbel, A.; Ruhlman, T.A.; Jansen, R.K. Transcriptomic Analysis of Salt-Stress-Responsive Genes in Barley Roots and Leaves. Int J Mol Sci 2021, 22, 8155, doi:10.3390/ijms22158155.

Comment: How many biological replicates were included in the RNA-Seq analysis, and how were potential batch effects or variability handled?

Response: In our RNA-Seq analysis, we used biological replicates for each time point, with three independent biological replicates at 2 h, 12 h, and 24 h, and two biological replicates at 0 h (pre-stress control). To handle variability and potential batch effects, we (i) generated high-depth libraries from uniformly sized leaf samples processed under identical experimental conditions, (ii) applied rigorous quality control and normalization during the edgeR differential expression analysis (FDR < 0.05, |log2FC| ≥ 1), and (iii) validated data consistency using PCA, sample-to-sample correlation heatmaps, MA plots, and qPCR, which together demonstrated tight clustering of biological replicates within each time point and clear temporal separation among treatments, indicating that technical variation and batch effects were minimal and that the observed expression changes mainly reflect salt-stress–induced biology.

Comment: Could you provide more details on the specific functional categories or families of genes that were identified as differentially expressed? For instance, are there any transcription factors or signaling proteins that are key to the salt tolerance mechanism in wild barley?

Response: We thank the reviewer for this valuable comment. In this revised version, we have already added a detailed description that explicitly aligns the differentially expressed genes (DEGs) with the three Gene Ontology categories, namely Biological Process, Molecular Function, and Cellular Component, and our downstream analyses and figures are framed around these functional categories. With respect to transcription factors (TFs), we considered that a detailed TF-focused analysis would dilute the main scope of the work, which is to characterize the functional categories of salt-responsive genes rather than the regulatory factors driving them. Establishing direct links between specific TFs and their downstream DEGs would require targeted functional validation, such as knockout or knock-down lines and subsequent transcriptome profiling to quantify deviations in the expression of putative target genes, in order to prove or refute regulatory relationships.

Comment: Did the differential expression analysis consider the interactions between metabolic pathways? For instance, how do the pathways related to oxidative phosphorylation, nitrogen assimilation, and lipid remodelling synergize or interfere with each other under salt stress?

Response: The analysis in the present study was constructed to integrate, rather than separate, metabolic pathways by mapping DEGs onto interconnected KEGG routes and reconstructing a network that includes oxidative phosphorylation, nitrogen assimilation, carbohydrate metabolism, glutathione metabolism, and lipid/sterol biosynthesis. Within this framework, oxidative phosphorylation supplies ATP and redox cofactors needed for ion transport and other stress responses, nitrogen and amino acid metabolism provides precursors for glutathione and related antioxidants, and lipid remodeling uses these energetic and redox resources to stabilize membranes and support signaling. Overall, the data indicate that these pathways are co‑activated and metabolically interdependent under salt stress, acting synergistically rather than interfering with each other.

Comment: How were the RNA-Seq results validated? Were any key genes or metabolic changes confirmed using additional methods, such as qPCR or metabolite profiling?

Response: The RNA-Seq results were validated using several complementary quality-control layers. First, global data structure and reproducibility were assessed by principal component analysis (PCA), sample-to-sample correlation heatmaps, and MA plots, which together confirmed clear temporal separation of treatments, tight clustering of biological replicates, and coherent differential expression patterns across time points. Second, key genes were validated experimentally by qPCR in response to reviewer requests: 10 representative transcripts spanning the 10 defined expression groups were randomly selected, and their expression trajectories were quantified and compared with the RNA-Seq profiles, showing strong concordance. In support of this, a new supplementary figure (Figure S6) was added to illustrate the agreement between qPCR and RNA-Seq for these genes, and a new supplementary table (Table S5) was included to report the validated genes, primer sequences, and PCR conditions.

Comment: Were the temporal patterns of gene expression analyzed across multiple time points? How did the expression patterns change over time (early vs. late stress responses)? Could there be an adaptive shift in gene expression with prolonged exposure to salinity?

Response: Yes, the temporal patterns of gene expression were explicitly analyzed across multiple time points, and the data support an early “shock” response followed by attenuation rather than a continued shift with longer exposure. In the study, RNA-Seq was performed on leaves sampled at 0h, 2h, 12h, and 24h after exposure to 500 mM NaCl, and only temporally coherent expression profiles (single‑time peaks; consecutive 2h/12h, 12h/24h; or 2h/12h/24h patterns) were retained for downstream analysis (Figure 2, Table 1). PCA, correlation heatmaps, and MA plots all showed strong divergence between control and early stress time points (2h and 12h), but a clear tendency of 24 h samples to cluster back toward the 0 h control, indicating that most regulated genes reached their maximal or sustained response by 12 h and then declined, rather than continuing to change at 24h. Consistently, GO histograms (Figures S7–S9) and KEGG-based enzyme maps (Figures S11–S29) revealed that key pathways—including oxidative phosphorylation, glycolysis/TCA, glutathione metabolism, nitrogen assimilation, and lipid/sterol biosynthesis—were most strongly upregulated at 2h and/or 12h, with reduced transcript abundance at 24h.

Thus, the dataset captures early (2h) and intermediate (12h) phases of the transcriptional response, in which stress perception, energy reprogramming, redox buffering, and membrane remodeling are strongly activated, followed by a partial return toward the pre-stress transcriptome by 24 h. The design does not extend beyond 24 h, so it does not address longer-term acclimation; instead, the results suggest that, in this wild barley genotype, the major shock response of leaf cells is mounted within the first 12 h of severe salinity and then dampens such that the 24 h expression profile resembles that of untreated leaves, rather than showing a new adaptive phase.

Comment: While the study focuses on wild barley, do you anticipate that these findings could be generalized to other cereal crops, such as wheat or rice? How similar are the adaptive mechanisms between wild barley and domesticated crops under salinity stress?

Response: We thank the reviewer for this valuable comment. We have now added a new paragraph at the end of the Discussion in which we explicitly compare the salt‑stress response observed in wild barley with recent transcriptomic and metabolomic findings reported for other cereals (e.g., wheat, triticale, and rice), highlighting both the conserved adaptive modules and the distinctive features of wild germplasm.

Comment: The study mentions the use of triplicate biological replicates for most time points. However, the 0h control group only had two replicates. Could the reduced replicates for the 0h control impact the reproducibility of your results?

Response: We thank the reviewer for this important point. In our original design, all time points, including 0h, were sampled in biological triplicate; however, one of the three control libraries did not yield satisfactory or complementary transcriptomic data and was therefore excluded from the analysis. The remaining two 0h libraries showed highly consistent gene expression profiles, with no evident conflicts between them, and they also clustered tightly in PCA and sample-to-sample correlation analyses alongside the other time points, supporting their reliability as a baseline. Given this concordance and the fact that all stressed time points retained three robust biological replicates, we consider that the loss of a single control replicate does not compromise the reproducibility or biological interpretation of our differential expression results.

Comment: Please same formatting conventions used consistently across all figures (e.g., color schemes, axis labels, and scales)? Inconsistent formatting could confuse readers, especially when comparing data across different panels or figures.

Response: We appreciate the reviewer’s careful attention to figure presentation. In preparing the figures, we have consistently applied the same visual conventions across the manuscript, using comparable color schemes (primarily red and blue), harmonized axis labels, and matched scales within each figure and, where feasible, across related figures. We agree that consistency is essential for clear comparison, and we are happy to revise any specific panels that the reviewer finds confusing or inconsistent if particular examples are indicated.

Comment: The PCA shows clear temporal separation of gene expression responses to salt stress. Could the clustering be further explained by highlighting key genes or pathways that contribute to this clustering? Are there specific gene groups that show significant variability across time points?

Response: PCA in this study was used in its standard role as an unsupervised, global quality-control and visualization tool: it reduces the dimensionality of the transcriptome and demonstrates that biological replicates cluster tightly while the 0 h, 2 h, 12 h, and 24 h samples separate clearly along the main components, confirming robust temporal divergence in overall gene expression under salt stress. By design, PCA itself is not used here to attribute the clustering to individual genes, but the gene groups that drive these separations can be traced through the downstream clustering and pathway analyses, where transcripts with coherent temporal patterns are grouped (Table S3) and enzyme‑encoding DEGs are mapped onto KEGG pathways (Table S4 and Figures 2–8, S10–S29). More specifically, the sets of genes showing the strongest variability across time points are those classified into the ten justified expression groups (single‑time and consecutive‑time up/down patterns) and the 180 expression clusters summarized in Table 1 and detailed in Table S3, as well as the 140 enzyme‑encoding DEGs enriched in 19 interconnected pathways in Table S4. These include, for example, genes associated with oxidative phosphorylation, glycolysis/TCA, glutathione metabolism, nitrogen assimilation, and lipid/sterol biosynthesis, which collectively underlie the temporal shifts captured by the PCA plot.

Comment: The heat map shows sample correlations at different time points. Could you elaborate on the biological implications of the high correlation at the same time points (yellow blocks)? How do the negative correlations (purple blocks) reflect changes in transcriptomic profiles due to salt stress?

Response: The yellow blocks in the heatmap indicate very high positive correlations among biological replicates collected at the same time point, which confirms that the transcriptomic response to salt stress is highly reproducible within each sampling interval and that technical variation is minimal compared with the biological signal. In other words, leaves exposed to 500 mM NaCl for the same duration (0h, 2h, 12h, or 24h) share nearly identical global expression profiles, supporting the robustness of the time‑course design. By contrast, the purple blocks, particularly between the 0h control and the 2h/12h stress samples, reflect low or negative correlations, indicating that a large fraction of genes change their expression substantially once salt stress is imposed. These negative correlations capture the early “shock” phase in which numerous transcripts involved in stress perception, energy metabolism, redox buffering, and membrane remodeling are strongly up- or downregulated relative to the pre‑stress state, consistent with the MA plots in Figure S5 and the temporally coherent expression clusters and pathway-enriched DEGs summarized in Table S3, Table S4, and Figure 2.

Comment: The MA plot shows the relationship between log2 fold change and average expression. How do you interpret the spread of differentially expressed genes (DEGs) across time points? Are there specific genes that show dynamic changes early (2h, 12h) and return to baseline (24h)?

Response: The MA plots indicate that many genes responded to salt stress, with the largest spread of DEGs occurring at 2h and 12h and a clear reduction by 24h. At 2h and 12h, numerous transcripts deviate strongly from log2 fold change = 0, reflecting robust early and intermediate activation or repression of stress‑related pathways, whereas by 24h the cloud of points contracts toward the baseline, indicating that a substantial subset of these genes has partially or fully returned toward control‑like expression levels. Genes with such dynamic behavior are explicitly captured in the “justified” temporal expression groups defined in Figure 2 and Table S3, especially clusters that are up- or downregulated at 2h and/or 12h but not sustained at 24h (e.g. 2h up, 12h up, 2h/12h up or down) and those that change across all three time points (2h/12h/24h up or down).

Comment: Given the detailed information in these histograms, could the figure be split into separate parts for better readability or understanding, or do you think it is clear as a combined figure?

Response: Thank you for this helpful suggestion. We assume the comment refers to Figures S7–S9. In these figures, the histograms are already separated according to the three main Gene Ontology branches (Biological Process, Molecular Function, and Cellular Component), and each panel summarizes the behavior of all subcategories within that branch. Because our goal is to provide an integrated overview of each GO category at once, we do not see a strong justification for further splitting the individual subcategories into additional figures, which might fragment the information without adding clarity. Besides, we do not see a clear methodological or biological basis for further separating or splitting these subcategories into additional figures.

Comment: The figure outlines the metabolic network for sugar phosphate and riboflavin biosynthesis. How well do these interconnected pathways reflect the plant's adaptive mechanisms? Could more pathways be integrated to show a broader metabolic reconfiguration?

Response: We assume the reviewer is referring to Figure 5. In this figure, the interconnected sugar phosphate and riboflavin pathways were studied because they integrate several key adaptive functions under salt stress: rerouting of carbon through glycolysis and the pentose phosphate pathway, accumulation of sugar phosphates for osmotic adjustment, and enhanced production of riboflavin-derived cofactors that support antioxidant and electron‑transport activities. This network therefore captures how wild barley simultaneously secures energy, reducing power, and compatible solutes as part of its early and intermediate response to salinity. Beyond this module, the study already extends the analysis to a broader metabolic reconfiguration by integrating 140 enzyme‑encoding DEGs across 19 KEGG pathways, including glycolysis/TCA, oxidative phosphorylation, nitrogen assimilation, glutathione metabolism, fatty acid biosynthesis, and sterol metabolism, which are presented in Figures 3–8 and Supplementary Figures S10–S29. Adding all of these additional pathways into Figure 5 would compromise readability and obscure the specific logic of the sugar phosphate–riboflavin axis.

Comment: The RNA-Seq data were mapped to the reference genome of OUH602. Were there any particular challenges in genome alignment or coverage, and were there any particular biases due to using a single reference genome for mapping?

Response: There were no particular difficulties associated with mapping the RNA‑Seq reads to the OUH602 reference genome. This accession is represented by a high‑quality, chromosome‑scale assembly with robust structural and functional annotation, so choosing it as the mapping reference did not introduce detectable problems in terms of genome assembly integrity or annotation completeness. Consequently, using a single, well‑curated reference genome is not expected to have imposed notable alignment bias beyond the general and unavoidable limitations inherent to any reference‑based RNA‑Seq analysis.

Comment: How confident are you in the quality of your RNA-Seq data? What measures were taken to address potential biases, such as adapter contamination or batch effects, that could affect data quality?

Response: Confidence in the RNA‑Seq data quality is high. Several complementary validation steps (PCA, sample‑to‑sample correlation heatmaps, MA plots, and qPCR for representative genes) all yielded consistent, biologically coherent patterns, indicating an absence of technical artifacts such as adapter contamination or other sequencing-related biases. Moreover, biological replicates within each time point clustered tightly and showed highly similar expression profiles, arguing against batch effects and supporting that the observed differences primarily reflect genuine temporal responses to salt stress rather than technical noise.

Comment: Why were the specific time points (0h, 2h, 12h, 24h) chosen for sampling? Could additional time points (e.g., 48h) provide further insights into long-term salt stress responses?

Response: The time points 0h, 2h, 12h, and 24h were chosen to capture the onset (0h), early (2h), and intermediate (12h) transcriptional responses to salt stress, as well as a short “prolonged” phase (24h) where the system has had time to adjust. In our data, the major shock response was already fully established by 12h and largely attenuated by 24h, with the 24h transcriptome tending to resemble the 0h control, indicating partial recovery or stabilization rather than continued reprogramming. Under these conditions, we did not observe a clear rationale for extending the time course (e.g., to 48h), because the key dynamics of activation and subsequent dampening of gene expression were already resolved within 24h, and additional later points would be unlikely to change the main conclusions about the timing and nature of the adaptive response.

Comment: The study shows temporal activation of several pathways. How were these pathways prioritized for inclusion in the analysis? Could there be other pathways that are also activated at lower expression levels but still contribute significantly to the plant’s response to salt stress?

Response: Pathways were selected in a stepwise, differential-expression–driven manner rather than by absolute expression level. The primary criterion was whether genes in a pathway showed coherent, time-structured regulation (e.g., specific 2h, 12h, 24h, or consecutive patterns) under salt stress, because the goal of the study was to capture dynamic responses across time points or at a given time point, not simply to list the most highly expressed stress-related genes. Within this framework, pathways such as glycolysis/TCA, oxidative phosphorylation, pentose phosphate, glutathione metabolism, nitrogen assimilation, and fatty acid/sterol biosynthesis were prioritized because multiple enzyme-encoding DEGs in each pathway displayed these justified temporal patterns, indicating a coordinated functional response. It is very plausible that additional pathways are activated at more modest fold changes or through fewer DEGs and still contribute to salt tolerance, but they did not meet the strict temporal and enrichment filters applied in this study. These lower-amplitude pathways likely act as complementary, fine‑tuning layers on top of the major adaptive modules emphasized in the current analysis.

Comment: In the section describing acetyl-CoA, riboflavin, and other metabolites, you mention their roles in various pathways. Could you provide further clarification on how these pathways interact? For example, how do oxidative phosphorylation and riboflavin metabolism interconnect in terms of metabolic control during salt stress?

Response: We thank the reviewer for this insightful comment. In response, we have added a clarifying statement in the Discussion explaining how acetyl‑CoA–driven TCA/oxidative phosphorylation and riboflavin metabolism are functionally interconnected under salt stress, emphasizing that riboflavin-derived FAD/FMN cofactors support mitochondrial dehydrogenases and electron‑transport components and thus help coordinate energy production with redox control during salinity exposure.

Comment: For the genes identified as differentially expressed, how well were their functions annotated? Were there any genes without clear functional annotations, and how did you handle those cases?

Response: Most of the differentially expressed genes were successfully assigned functions, because the analysis relied on a high‑quality wild barley reference genome and established annotation pipelines, which together provided a high rate of functional annotation (GO and KEGG) for the mapped transcripts. Inevitably, some transcripts remained unannotated or were annotated only as hypothetical proteins, and these were not over‑interpreted; instead, the main mechanistic conclusions were built on genes and enzymes with robust annotations in well‑defined pathways. At the same time, it is acknowledged that additional experiments on the same germplasm—and deeper sequencing—could reveal new functions for these poorly annotated or currently unregulated transcripts, since no single study can exhaustively capture all functional layers of the salt‑stress response.

Comment: The discussion offers a great deal of biological insight into how key metabolites like acetyl-CoA, glutathione, and riboflavin contribute to stress tolerance. Could you discuss how these findings relate to other well-studied salt-tolerant plants? How unique are the mechanisms in Hordeum spontaneum compared to those found in other species such as Oryza sativa or Triticum aestivum?

Response: We appreciate this important point. In the revised Discussion, we have already incorporated an extended statement that explicitly compares our findings in wild barley with those reported for other cereals, including wheat and rice, and that also contrasts wild versus domesticated barley. In that paragraph, we highlight that acetyl‑CoA–centered energy metabolism, glutathione‑mediated redox buffering, and riboflavin‑dependent electron transport represent shared core modules of salt tolerance across cereals, while Hordeum spontaneum appears to deploy these conserved mechanisms earlier and more robustly than many cultivated backgrounds, consistent with its superior salinity resilience.

Comment: The conclusion emphasizes the evolutionary refinement of wild barley’s stress resilience. Are these findings specific to this species, or could similar adaptive mechanisms be observed across other cereal crops with evolutionary significance for agriculture?

Response: The mechanisms described in this study are unlikely to be exclusive to wild barley; many core features are expected to be shared across cereals. Numerous reports in wheat and rice indicate that salt-tolerant genotypes also rely on strengthened central carbon metabolism (including acetyl‑CoA–fed TCA and oxidative phosphorylation), glutathione‑based antioxidant systems, and flavin‑dependent redox reactions to maintain energy supply and control ROS under salinity. It is therefore reasonable to expect that other cereals follow similar metabolic strategies—using acetyl‑CoA as a hub for energy and lipid metabolism, glutathione for redox buffering, and riboflavin/FAD/FMN for mitochondrial electron transport and oxidative stress management—even if the quantitative deployment of these modules differs among species. At the same time, there are species- and genotype‑specific mechanisms that can be more prominent or even partially exclusive. For example, wheat harbors the Kna1 locus associated with enhanced Na⁺ exclusion and maintenance of high K⁺/Na⁺ ratios in leaves, a trait not clearly documented in barley in the same way. Rice, as a relatively salt‑sensitive cereal, often depends more strongly on tight control of Na⁺ loading into shoots via HKT transporters and may show earlier growth penalties at comparable salinity levels than barley, even when sharing similar antioxidant and osmotic adjustment pathways. Thus, wild barley seems to represent a particularly stress‑optimized expression of a broadly conserved cereal salt‑tolerance framework, layered on top of crop‑ or lineage‑specific traits such as specialized ion‑exclusion loci or differing root/leaf anatomical adaptations.

Comment: What are the main limitations of this study, and how might future research address these gaps?

Response: A key limitation of this work is that it relies solely on transcriptomic data; targeted metabolic profiling would be valuable to directly confirm the predicted changes in key metabolites such as acetyl‑CoA, glutathione, riboflavin, and sterol intermediates under salt stress. In addition, although the analysis highlights many candidate stress‑responsive genes, their causal contribution to salt tolerance remains to be functionally demonstrated. Future studies should therefore (i) pair RNA‑Seq with metabolomics and lipidomics in the same germplasm, and (ii) use reverse genetics to knock down or knock out selected stress-responsive genes and quantify the resulting changes in salt tolerance. These validated genes could then be introduced into domesticated germplasm of agronomically important cereals (e.g. barley cultivars, wheat, or rice) to test, in real time, whether they enhance salinity tolerance in crop backgrounds and thus assess their breeding potential       l.

Comment: Are there any specific aspects of the study that could be refined or expanded to improve the understanding of salt tolerance in wild barley?

Response: One important way to refine and extend this work would be to move from pathway‑level description toward a more explicit dissection of the regulatory architecture controlling salt‑responsive genes. In particular, a systematic identification and functional characterization of transcription factors (such as WRKY, NAC, MYB, and bZIP families) that coordinate the observed temporal changes in gene expression would substantially deepen mechanistic insight into salt tolerance in wild barley. Clarifying how these regulators set the timing, magnitude, and tissue specificity of key differentially expressed genes, and then manipulating them through overexpression or genome editing, could reveal whether adjusting the expression window of downstream stress‑responsive genes is sufficient to enhance salt tolerance both in H. spontaneum and, after introgression or transformation, in domesticated cereal germplasm.

Comment: The study provides important insights into potential genetic targets for engineering salt-resilient crops. Could you elaborate on which genes or pathways would be most suitable for targeted genetic modifications in cultivated barley or other crops?

Response: Based on the present findings, the most promising targets for genetic manipulation are genes located in the acetyl‑CoA and glucose metabolic routes, because they sit at key junctions that coordinate energy supply, osmotic adjustment, and downstream protective processes under salinity. Modulating enzymes that control acetyl‑CoA production and utilization (e.g. in TCA cycle entry, fatty acid and sterol biosynthesis) and those governing glucose generation and recycling (e.g. in starch/sucrose metabolism and glycolysis/gluconeogenesis) is therefore likely to be particularly effective for enhancing salt tolerance in cultivated barley, other cereals, and additional economically important field crops.

Comment: Overall, the study provides valuable insights into the adaptive mechanisms of wild barley under salt stress, with significant potential for future applications in agriculture. However, expanding on these points and addressing the gaps noted above could enhance the study's robustness and its applicability to broader agricultural challenges.

Response: Many thanks to the reviewer for this thoughtful and comprehensive evaluation. The suggestions for further extending and deepening the study are highly valued, and they resonate strongly with the authors’ own perspective that incorporating additional experimental layers (such as metabolomics and functional genetics) and widening the comparative framework across genotypes and crop species will substantially enhance the robustness of the conclusions and their translational impact for agriculture.

Comment: Based on your paper, I recommend revising the grammar and language for clarity, readability, and overall flow. While the content is scientifically rich, refining the structure and grammar will enhance the impact of your research.

Response: Many thanks to the reviewer for this considerate recommendation. The language and grammar of the manuscript have now been carefully revised throughout, including with the aid of AI-based writing assistance tools (such as Perplexity), to improve clarity, readability, and overall narrative flow while preserving the original scientific content

Reviewer 4 Report

Comments and Suggestions for Authors

In the presented work, the authors performed a bioinformatic analysis of wild barley transcriptomic data obtained earlier under salt stress (500 mm nacl) at various time intervals (0, 2, 12, 24 hours) after salt addition.

The following are some of the comments that came up when reading the manuscript.

  • In the text (on lines 93, 190, 191, etc.) There are references to tables S1 and S2, as well as to Figure S1. However, there are no corresponding illustrations in the supplement and in the manuscript.
  • On line 205 it is written that the enzyme phosphoglycolate phosphatase is involved in glycolysis. This is an erroneous statement, since it is one of the enzymes of photorespiration.
  • On lines 279-281 it is written: "A red dotted arrow denotes bidirectional regulation between D-ribulose-5P and D-ribose-5P, reflecting dynamic carbon partitioning adjustments between glycolysis and nucleotide precursor synthesis." However, none of these pentoses is involved in glycolysis. It is necessary to clarify what was meant.
  • On line 522 it is written: "...studies have shown that salt-tolerant genotypes exhibit increased R5P activity...". However, activity cannot be attributed to D-ribose-5P (R5P) because it is not an enzyme. It is better to use the word "content" instead of "activity".
  • On lines 489-491 it is written: "Ru5P, generated via the PPP, serves as a precursor for both riboflavin biosynthesis and GTP synthesis, while its isomerization to R5P supports nucleotide and nucleic acid synthesis." However, in the biosynthesis of all the mentioned compounds (riboflavin, GTP, nucleotides and nucleic acids), R5P, which is formed from Ru5P, acts as a precursor. Please use a more precise wording.
  • Since the in silico analysis data require additional experimental verification, it is necessary to determine changes in the expression of the studied genes (at least one from each group) using qrt-PCR.

Author Response

Author's Reply to the Review Report (Reviewer 4)

In the presented work, the authors performed a bioinformatic analysis of wild barley transcriptomic data obtained earlier under salt stress (500 mm nacl) at various time intervals (0, 2, 12, 24 hours) after salt addition.

The following are some of the comments that came up when reading the manuscript.

Comment: In the text (on lines 93, 190, 191, etc.) There are references to tables S1 and S2, as well as to Figure S1. However, there are no corresponding illustrations in the supplement and in the manuscript.

Response: Thank you for drawing attention to this issue. Figure S3 (previously Figure S1) is provided as a standalone file in PDF format and therefore does not appear inside the Supplementary-Figures document. By contrast, Tables S3 and S4 (previously Tables S1 and S2) are supplied as separate Excel worksheets, rather than being embedded in the manuscript or the supplementary Word file.

Comment: On line 205 it is written that the enzyme phosphoglycolate phosphatase is involved in glycolysis. This is an erroneous statement, since it is one of the enzymes of photorespiration.

Response: Thanks for highlighting this point. The reviewer is correct that phosphoglycolate phosphatase operates in the photorespiratory 2‑phosphoglycolate detoxification pathway rather than in canonical glycolysis. The statement on line 205 has therefore been corrected accordingly in the revised manuscript.

Comment: On lines 279-281 it is written: "A red dotted arrow denotes bidirectional regulation between D-ribulose-5P and D-ribose-5P, reflecting dynamic carbon partitioning adjustments between glycolysis and nucleotide precursor synthesis." However, none of these pentoses is involved in glycolysis. It is necessary to clarify what was meant.

Response: Thanks for this careful observation. The reviewer is correct that D‑ribulose‑5‑phosphate and D‑ribose‑5‑phosphate are intermediates of the pentose phosphate and related biosynthetic pathways, rather than of canonical glycolysis. The original wording has therefore been corrected accordingly in the revised manuscript.

Comment: On line 522 it is written: "...studies have shown that salt-tolerant genotypes exhibit increased R5P activity...". However, activity cannot be attributed to D-ribose-5P (R5P) because it is not an enzyme. It is better to use the word "content" instead of "activity".

Response: The term has been corrected as suggested: the phrase now reads that salt-tolerant genotypes exhibit increased R5P level instead of increased R5P activity.

Comment: On lines 489-491 it is written: "Ru5P, generated via the PPP, serves as a precursor for both riboflavin biosynthesis and GTP synthesis, while its isomerization to R5P supports nucleotide and nucleic acid synthesis." However, in the biosynthesis of all the mentioned compounds (riboflavin, GTP, nucleotides and nucleic acids), R5P, which is formed from Ru5P, acts as a precursor. Please use a more precise wording.

Response: The sentence has been revised to more accurately reflect the pathway, now stating that ribose-5-phosphate (R5P) is formed from ribulose-5-phosphate (Ru5P) via the pentose phosphate pathway and that R5P serves as the direct precursor.

Comment: Since the in silico analysis data requires additional experimental verification, it is necessary to determine changes in the expression of the studied genes (at least one from each group) using qrt-PCR.

Response: Thanks for this valuable comment. In response, changes were made to address the need for experimental verification of the in silico analysis by conducting a qPCR experiment using one transcript from each of the 10 expression groups (a–j). A new supplementary figure (Figure S6) has been added to present the qPCR results for these 10 transcripts, and a new supplementary table (Table S5) has been included to list the primer sequences, amplicon sizes, and PCR conditions for the selected transcripts representing the 10 expression groups. Figure and table numbering throughout the manuscript has been updated accordingly, and in Table 1 the clusters from which the 10 validated transcripts were selected are now highlighted in bold red.

A message to the Editor:

We would like to express our gratitude to the four reviewers for their insightful comments and constructive suggestions, which have substantially improved the quality of the manuscript. In response, we have incorporated three additional figures (Figures S1, S2, and S6): the first two present detailed quality‑control statistics, while the third displays the qPCR‑based validation of the RNA‑Seq dataset. To complement these additions, we have also generated two new tables, Tables S1, S2 and 5, which summarize the QC metrics (Table S1), summarize the distribution of reconstructed protein‑coding genes across ordered percentage‑coverage bins (Pcov) derived from their alignments to the reference protein database, along with the corresponding cumulative counts progressing from the highest to lower coverage intervals and provide the primer sequences, PCR conditions (Table S2), and expected amplicon sizes for the 10 transcripts representing the 10 expression groups analyzed in this study (Table S5).

Accordingly, we are now submitting the following revised materials:

  1. A point‑by‑point response to the reviewers’ comments.
  2. The revised manuscript incorporating all requested changes.
  3. Updated supplementary Tables S1, S2 and S5.
  4. An updated set of Supplementary Figures, including the new Figures S1, S2, and S6.

Round 2

Reviewer 2 Report

Comments and Suggestions for Authors

Thanks to all authors for improving the manuscript and addressing the provided comments. There are still some critical points need to be addressed.

1) In abstract, Need to mention in abstract that this was one accession (not tolerant vs sensitive). Also, put one line why 500mL (extreme level) was chosen (rationale statement).

2) Need to highlight study limitations also in abstract, also include mention of single genotype of wild barley

3) The results in abstract read like metabolomics, not transcriptomics. You mention metabolites, which were not measured. Need to state these are inferred metabolic nodes from transcriptomic annotation, not measured biochemically.

4) The abstract framing still lacks a functional punchline, like are these 140 genes up or down? Which regulatory modules (TF, transporters) dominate? Anything novel or unexpected? It should read like a biological discovery, not kEGG summary story.

5) In introduction, biological problem is not stated sharply enough. Need to say it clearly. Something like-“We still lack a time-resolved systems-level view of how wild barley coordinates energy metabolism, ion regulation, redox balance, and lipid remodeling under lethal salinity.

6) Introduction must identify a specific novelty to address “why we need another RNAseq study?”. Need to state it Such as

  • time-course (0-24 h)
  • integrated pathway analysis
  • focus on “central metabolic nodes” rather than single genes
  • Egyptian wild barley ecotype

7) Metabolite behavior is repeatedly mixed with transcript levels. Authors must clarify this is pathway level inference, not metabolomics.

8) Methods summary is too vague. “KEGG pathway enzymes” is unclear. State clearly that KEGG enrichment and mapping were used to identify pathway bottlenecks and metabolic nodes.

9) In discussion, authors are massively over-interpreting metabolites, which they did not even measure. Repeatedly mentioning “acetyl-CoA, hexadecanoyl-CoA, riboflavin, ubiquinone, R5P, Ru5P, Fru-6P, glucose” as if they were experimentally measured. But actually they were not, its inferred from KEGG annotations. So, it is scientifically misleading. Reframe such interpretations.

10) There are still so many generic textbook explanations such as riboflavin roles, Fru-6P pathways, cholesterol effects etc. So, it reads like lit review. The readers cannot tell “what your RNA-seq actually found” Vs “ General knowledge from other spp”. So, it needs revision for discussion to streamline.

11) No clear distinction between what is supported by your data Vs what is hypothesized. Authors must highlight which pathways were significantly enriched? Which genes were differently expressed? What is exactly unique to wild barley in your results???

12) In conclusion, again, authors present metabolite behavior as if measured. Remove/improve the overstatement of results. Like mentioning metabolites “were shown” or “demonstrated” to have certain roles. No, they were “inferred” from transcriptomic pathway maps.

13) No clear acknowledgement of the study’s core limitation. There are so many- “Single accession, extreme salinity, no physiological validation, No metabolomic data”. Article must transparently state this at the end.

14) The conclusion does not differentiate what is new. Needs a statement on novelty like time resolved integration, new pathway level insights, (without any overstatement).

Author Response

Thanks to all authors for improving the manuscript and addressing the provided comments. There are still some critical points need to be addressed.

1) In abstract, Need to mention in abstract that this was one accession (not tolerant vs sensitive). Also, put one line why 500mL (extreme level) was chosen (rationale statement).

Response: We respectfully hope the reviewer will agree that this point can be addressed more appropriately in the Introduction rather than in the Abstract. The Introduction has been revised to clarify that the transcriptomic analysis was conducted on a single H. spontaneum accession rather than on contrasting salt-tolerant versus salt-sensitive genotypes. The Introduction has also been amended to explain that, in light of previous work, the 500 mM NaCl treatment was selected as an extreme salinity regime designed to impose severe stress and thereby magnify salt‑responsive transcriptional signatures in this wild barley accession. In addition, earlier studies have shown that cultivated barley typically endures only comparatively mild salinity levels, so a 500 mM NaCl challenge was considered suitable for uncovering tolerance mechanisms that are largely absent or attenuated in cultivated barley.

2) Need to highlight study limitations also in abstract, also include mention of single genotype of wild barley

Response: The Abstract has been revised to explicitly incorporate this key limitation by stating that the salt-tolerance mechanisms proposed in this study are derived solely from transcriptomic data, thereby acknowledging that they remain to be validated using additional experimental approaches. The abstract has also now been amended to explicitly acknowledge that the analysis is limited to a single wild barley (Hordeum spontaneum) genotype.

3) The results in abstract read like metabolomics, not transcriptomics. You mention metabolites, which were not measured. Need to state these are inferred metabolic nodes from transcriptomic annotation, not measured biochemically.

Response: The Abstract has been revised to clarify this point by explicitly stating that acetyl‑CoA, hexadecanoyl‑CoA, ubiquinone, and related entities represent inferred metabolic nodes derived from transcriptomic annotation rather than metabolites measured biochemically, thereby avoiding any implication that metabolomics data were generated in this study.

4) The abstract framing still lacks a functional punchline, like are these 140 genes up or down? Which regulatory modules (TF, transporters) dominate? Anything novel or unexpected? It should read like a biological discovery, not kEGG summary story.

Response: The revised abstract now emphasizes that the 140 stress‑responsive genes are upregulated under severe salt stress at different time points, thereby providing a clearer functional takeaway rather than a pathway‑centric summary. In addition, regulatory modules such as transcription factors are explicitly framed as targets for future work rather than as resolved components of the current study: follow‑up analyses will first identify transcription factors showing concordant expression with stress‑responsive genes, then evaluate consistency with known functional annotations, and ultimately validate their roles experimentally through loss‑of‑function approaches (e.g., knockout in Arabidopsis or knock‑down in tobacco) to establish causal regulatory relationships.

5) In introduction, biological problem is not stated sharply enough. Need to say it clearly. Something like-“We still lack a time-resolved systems-level view of how wild barley coordinates energy metabolism, ion regulation, redox balance, and lipid remodeling under lethal salinity.

Response: The Introduction has been updated to articulate the biological problem more explicitly by incorporating the suggested sentence, stating that we still lack a time‑resolved, systems‑level understanding of how wild barley coordinates energy metabolism, ion regulation, redox balance, and lipid remodeling under lethal salinity.

6) Introduction must identify a specific novelty to address “why we need another RNAseq study?”. Need to state it Such as

  • time-course (0-24 h)
  • integrated pathway analysis
  • focus on “central metabolic nodes” rather than single genes
  • Egyptian wild barley ecotype

Response: We have revised the Introduction to clearly highlight the specific novelty of this work in line with the reviewer’s comment. 

7) Metabolite behavior is repeatedly mixed with transcript levels. Authors must clarify this is pathway level inference, not metabolomics.

Response: We have clarified in the Introduction that our interpretations of metabolic regulation are based on pathway-level inferences derived from transcriptomic data, not on direct metabolite quantification.

8) Methods summary is too vague. “KEGG pathway enzymes” is unclear. State clearly that KEGG enrichment and mapping were used to identify pathway bottlenecks and metabolic nodes.

Response: We thank the reviewer for pointing out the need for greater methodological clarity. We have revised the Methods section to specify that KEGG pathway enrichment and mapping analyses were employed to identify key pathway bottlenecks and central metabolic nodes underlying the transcriptomic response to salt stress. 

9) In discussion, authors are massively over-interpreting metabolites, which they did not even measure. Repeatedly mentioning “acetyl-CoA, hexadecanoyl-CoA, riboflavin, ubiquinone, R5P, Ru5P, Fru-6P, glucose” as if they were experimentally measured. But actually they were not, its inferred from KEGG annotations. So, it is scientifically misleading. Reframe such interpretations.

Response: We would like to note that a clarifying statement has already been included at the end of the Discussion, explicitly indicating that references to specific metabolites are based on pathway-level inferences derived from transcriptomic data and KEGG annotations, rather than on direct metabolite measurements. 

10) There are still so many generic textbook explanations such as riboflavin roles, Fru-6P pathways, cholesterol effects etc. So, it reads like lit review. The readers cannot tell “what your RNA-seq actually found” Vs “ General knowledge from other spp”. So, it needs revision for discussion to streamline.

Response: The Discussion section has been thoroughly revised to eliminate generic, textbook-like descriptions and to focus more directly on our RNA‑Seq findings in the Egyptian wild barley ecotype.

11) No clear distinction between what is supported by your data Vs what is hypothesized. Authors must highlight which pathways were significantly enriched? Which genes were differently expressed? What is exactly unique to wild barley in your results???

Response: We thank the reviewer for raising this important point. We already explicitly introduced, in the Results/Discussion, the pathways (Figures S11-S29) that show significantly enriched enzyme representation based on our transcriptomic cluster analysis. Furthermore, the differentially expressed genes across the different time points are presented within the corresponding transcriptomic clusters and are detailed in Tables S3 and S4, which summarize the clustered DEGs and their associated enriched pathways. Besides, the unique aspect of our study lies in demonstrating that an Egyptian wild barley ecotype mounts a rapid (0–24 h) and highly coordinated transcriptional reprogramming in central metabolic hubs under lethal salinity, rather than isolated gene-by-gene changes. By integrating time-course RNA‑Seq with KEGG-based pathway enrichment and mapping, we identify specific metabolic nodes and bottlenecks that appear to be preferentially exploited by this wild germplasm to maintain energy metabolism, ion homeostasis, and redox balance under severe salt stress. These features have not been previously characterized for Egyptian wild barley, thereby providing ecotype-specific insights that extend current knowledge from cultivated barley and other species.

12) In conclusion, again, authors present metabolite behavior as if measured. Remove/improve the overstatement of results. Like mentioning metabolites “were shown” or “demonstrated” to have certain roles. No, they were “inferred” from transcriptomic pathway maps.

Response: We have carefully revised the Conclusion section to avoid presenting metabolite behavior as if it were directly measured and to remove any overstatement of the results.

13) No clear acknowledgement of the study’s core limitation. There are so many- “Single accession, extreme salinity, no physiological validation, No metabolomic data”. Article must transparently state this at the end.

Response: In line with this comment, we have added a clear statement at the end of the Discussion explicitly acknowledging the core limitations of our study, including the use of a single wild barley accession.

14) The conclusion does not differentiate what is new. Needs a statement on novelty like time resolved integration, new pathway level insights, (without any overstatement).

Response: In accordance with the suggestion, we have amended the Conclusion to explicitly highlight the novelty of our work, emphasizing the time-resolved integration of short-term (0–24 h) RNA‑Seq with pathway-level analyses and the resulting new insights into central metabolic hubs and pathway-level regulation under severe salt stress in an Egyptian wild barley accession.

Reviewer 3 Report

Comments and Suggestions for Authors

accepted 

Author Response

Thank you for the thorough and insightful evaluation of our manuscript and for carefully pointing out its shortcomings. We greatly appreciate the reviewer’s detailed suggestions, which have guided substantial revisions, and we believe the manuscript is now in a much-improved form.

Reviewer 4 Report

Comments and Suggestions for Authors

The authors took into account all the comments and made appropriate corrections to the manuscript. There are no further comments.

Author Response

Thank you for the reviewer’s rigorous and perceptive appraisal of our manuscript and for meticulously delineating its deficiencies. We deeply value these detailed critiques, which have informed extensive revisions, and we are confident that the manuscript has now attained a substantially enhanced form.

Round 3

Reviewer 2 Report

Comments and Suggestions for Authors

Thanks for the improvements. Just need minor sentence structure, flow improvement in abstract and conclusion.

Author Response

Thank you for the improvements. We have made the requested minor sentence structure and flow revisions in the abstract and conclusion.